# Class-wise Generalization Error:
# an Information-Theoretic analysis

**Firas Laakom**                                    *firas.laakom@kaust.edu.sa*
*Center of Excellence for Generative AI*
*KAUST, Saudi Arabia*

**Moncef Gabbouj**                                  *moncef.gabbouj@tuni.fi*
*Faculty of Information Technology and Communication Sciences*
*Tampere University, Finland*

**Jürgen Schmidhuber**                              *juergen.schmidhuber@kaust.edu.sa*
*Center of Excellence for Generative AI, KAUST, Saudi Arabia*
*The Swiss AI Lab, IDSIA, Switzerland*

**Yuheng Bu**                                       *buyuheng@ucsb.edu*
*Department of Computer Science*
*University of California, Santa Barbara, CA, USA*[*]

**Reviewed on OpenReview:** *https://openreview.net/forum?id=asW4VcDFpi*

## Abstract

Existing generalization theories for supervised learning typically take a holistic approach and provide bounds for the expected generalization over the whole data distribution, which implicitly assumes that the model generalizes similarly for all different classes. In practice, however, there are significant variations in generalization performance among different classes, which cannot be captured by the existing generalization bounds. In this work, we tackle this problem by theoretically studying the class-generalization error, which quantifies the generalization performance of the model for each individual class. We derive a novel information-theoretic bound for class-generalization error using the KL divergence, and we further obtain several tighter bounds using recent advances in conditional mutual information bound, which enables practical evaluation. We empirically validate our proposed bounds in various neural networks and show that they accurately capture the complex class-generalization behavior. Moreover, we demonstrate that the theoretical tools developed in this work can be applied in several other applications.

## 1 Introduction

Despite the considerable progress towards a theoretical foundation for neural networks (He & Tao, 2020), a comprehensive understanding of the generalization behavior of deep learning is still elusive (Zhang et al., 2016; 2021). Over the past decade, several approaches have been proposed to uncover and provide a theoretical understanding of the different facets of generalization (He & Tao, 2020; Kawaguchi et al., 2017; Hochreiter & Schmidhuber, 1997; Roberts et al., 2022). In particular, multiple tools have been used to characterize the expected generalization error of neural networks, such as VC dimension (Sontag et al., 1998; Harvey et al., 2017), algorithmic stability (Bousquet & Elisseeff, 2000; Hardt et al., 2016b), algorithmic robustness (Xu & Mannor, 2012; Kawaguchi et al., 2022), and information-theoretic measures (Xu & Raginsky, 2017; Steinke & Zakynthinou, 2020; Wang & Mao, 2023). However, relying solely on the analysis

---

[*]This work was conducted while Yuheng Bu was with the Department of Electrical and Computer Engineering at the University of Florida, Gainesville, FL, USA.

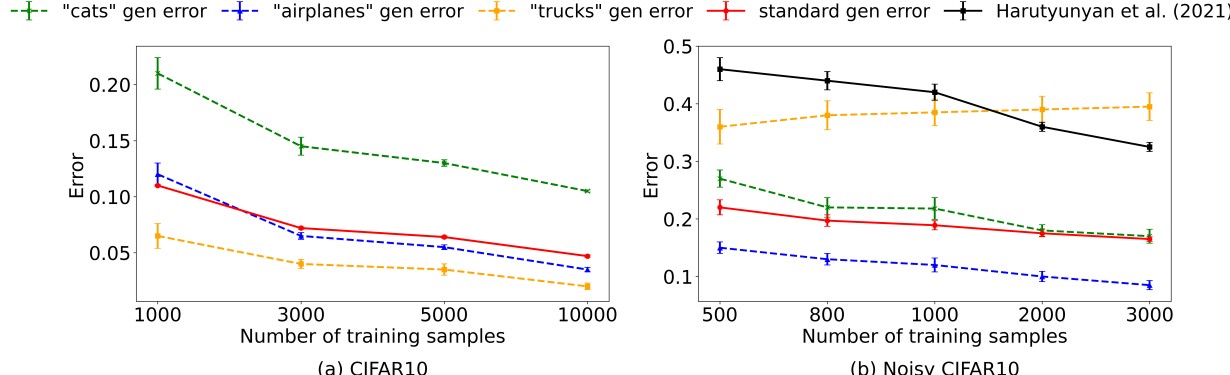

Figure 1: (a): The standard generalization error, i.e., test loss minus train loss, and the generalization errors for several classes on CIFAR10 as a function of the number of training samples. (b): The standard generalization error, bound proposed by Harutyunyan et al. (2021), and the generalization errors for several classes on noisy CIFAR10. By comparing both datasets, we can see that even minor corruptions in data distribution can lead to significant disparities in class-wise generalization behavior. It is worth-noting here that CIFAR10 is a balanced dataset, and results are averaged over $2 \times 20$ and $5 \times 15$ for CIFAR10 and its noisy variant, respectively. Experimental details are available in Section 4 and Appendix C.2.

of the expected generalization over the entire data distribution may not provide a complete picture. One fundamental limitation of the standard expected generalization error is that it does not give any insight into the class-specific generalization behavior, as it implicitly assumes that the model generalizes similarly for all the classes.

**Does the model generalize equally for all classes?** To answer this question, we conduct an experiment using deep neural networks, namely ResNet50 (He et al., 2016; Srivastava et al., 2015) on the CIFAR10 dataset (Krizhevsky et al., 2009). We plot the standard generalization error along with the class-generalization errors, i.e., the gap between the test error of the samples from the selected class and the corresponding training error, for three different classes of CIFAR10 in Figure 1 (a). As can be seen, there are significant variants in generalization performance among different classes. For instance, the model overfits the "cats" class, i.e., large generalization error, and generalizes relatively well for the "trucks" class, with a generalization error of the former class consistently four times larger than the latter. This suggests that *neural networks do not generalize equally for all classes*. Therefore, reasoning only concerning the standard generalization error (red curve), which averages over the classes, cannot capture this class-wise behavior.

Motivated by these observations, we conduct an additional experiment by introducing label noise (5%) to the CIFAR10 dataset to study how a slight change in data can affect the class-generalization behavior. Results are presented in Figure 1 (a). Intriguingly, despite the low noise level, the disparities between the class-wise generalization errors are aggravated, with some classes generalizing up to eight times worse than others. Further, as shown in this example, different classes do not even exhibit the same trend when the number of training samples increases. For instance, unlike the other classes, the generalization error of the "trucks" class increases when more training data is available. To further illustrate the issue of standard generalization analysis, we plot the information-theoretic generalization bound proposed in Harutyunyan et al. (2021). Although the bound captures the behavior of the standard generalization error well and can be used to explain the behavior of some classes (e.g., "cat"), it becomes an invalid upper bound for the "trucks" class[1].

When comparing the results on both datasets, it is worth noting that the generalization error of the same class "trucks" behaves significantly differently on the two datasets. This suggests that class-wise generalization highly depends on factors beyond the intrinsic class difficulty itself, including the data distribution, existence

---

[1]We note that the bound by Harutyunyan et al. (2021) is proposed for the standard generalization error instead of the class-generalization. Here, we plot it only for illustrative purposes.

of label noise, and the number of training samples. Moreover, in alignment with our findings, Balestriero et al. (2022); Kirichenko et al. (2023) showed that standard data augmentation and regularization techniques, e.g., weight decay and dropout (Krizhevsky et al., 2012; Hanson, 1990) improve standard average generalization. However, it is surprising to note that these techniques inadvertently increase the disparity of generalization among different classes.

The main conclusion of all the aforementioned observations is that *neural networks do not generalize equally for all classes, and their class-wise generalization depends on all ingredients of a supervised learning problem.* Furthermore, having more data may also exacerbate overfitting for certain classes. This paper aims to provide some theoretical understanding of this phenomenon using information-theoretic generalization bounds, as such bounds are both data-dependent and algorithm-dependent (Xu & Raginsky, 2017; Neu et al., 2021). This makes them an ideal tool to characterize the class-generalization properties of a learning algorithm. From a practical perspective, class-wise analysis becomes particularly relevant in scenarios involving high-stakes decisions tied to specific classes (e.g., medical diagnosis). In such cases, relying on averaged generalization errors can obscure critical performance gaps.

Our main contributions are as follows:

- We introduce the concept of "class-generalization error," which quantifies the generalization performance of each individual class. We derive a novel information-theoretic bound for this quantity based on KL divergence (Theorem 1). Then, using the super-sample technique proposed by Steinke & Zakynthinou (2020), we derive various tighter bounds based on conditional mutual information that are significantly easier to estimate and do not require access to the model's parameters (Theorems 2, 3, and 4). A visual overview is presented in Figure 2.

- We validate our proposed bounds empirically in different neural networks using CIFAR10 and its noisy variant in Section 4. We show that the proposed bounds can accurately capture the complex behavior of the class-generalization error behavior in different contexts.

- We show that our novel theoretical tools can be applied to the following cases beyond the class-generalization error: (i) Derive first class-dependent standard generalization error bounds highlighting how the class-generalization affects the standard generalization (Section 5.1) and in some cases tightening the existing standard generalization error bounds using class-dependency ; (ii) provide first practical tight bounds for the subtask problem, where the test data only encompasses a specific subset of the classes encountered during training (Section 5.2); (iii) derive generalization error bounds for learning in the presence of sensitive attributes (Section 5.3).

**Notations:** We use upper-case letters to denote random variables, e.g., $\mathbf{Z}$, and lower-case letters to denote the realization of random variables. $\mathbb{E}_{\mathbf{Z} \sim P}$ denotes the expectation of $\mathbf{Z}$ over a distribution $P$. Consider a pair of random variables $\mathbf{W}$ and $\mathbf{Z} = (\mathbf{X}, \mathbf{Y})$ with joint distribution $P_{\mathbf{W},\mathbf{Z}}$. Let $\overline{\mathbf{W}}$ be an independent copy of $\mathbf{W}$, and $\overline{\mathbf{Z}} = (\overline{\mathbf{X}}, \overline{\mathbf{Y}})$ be an independent copy of $\mathbf{Z}$, such that $P_{\overline{\mathbf{W}}, \overline{\mathbf{Z}}}(w, z) = P_{\mathbf{W}}(w) \cdot P_{\mathbf{Z}}(z)$. For random variables $\mathbf{X}$, $\mathbf{Y}$ and $\mathbf{Z}$, $I(\mathbf{X}; \mathbf{Y}) \triangleq D(P_{\mathbf{X},\mathbf{Y}} \| P_{\mathbf{X}} \otimes P_{\mathbf{Y}})$ denotes the mutual information (MI), and $I_z(\mathbf{X}; \mathbf{Y}) \triangleq D(P_{\mathbf{X},\mathbf{Y}|\mathbf{Z}=z} \| P_{\mathbf{X}|\mathbf{Z}=z} \otimes P_{\mathbf{Y}|\mathbf{Z}=z})$ denotes disintegrated conditional mutual information (CMI), which capture how much information $\mathbf{X}$ provides about $\mathbf{Y}$, not just globally, but conditioned on finer-grained instances of $\mathbf{Z}$. $\mathbb{E}_{\mathbf{Z}}[I_{\mathbf{Z}}(\mathbf{X}; \mathbf{Y})] = I(\mathbf{X}; \mathbf{Y}|\mathbf{Z})$ is the standard CMI. We will also use the notation $\mathbf{X}, \mathbf{Y}|z$ to simplify $\mathbf{X}, \mathbf{Y}|\mathbf{Z} = z$ when it is clear from the context.

## 2 Related Work

**Information-theoretic generalization error bounds:** Information-theoretic bounds have attracted a lot of attention recently to characterize the generalization of learning algorithms (Solomonoff, 1960; Neu et al., 2021; Wang et al., 2010; Aminian et al., 2021; Schmidhuber, 1997; Wu et al., 2020; Wang & Mao, 2022; Modak et al., 2021; Wang & Mao, 2021; Shui et al., 2020; Wang et al., 2023; Alabdulmohsin, 2020) in terms of information shared between the training data, and the learned hypothesis, providing insight into how data-dependent and algorithm-dependent interactions govern overfitting.

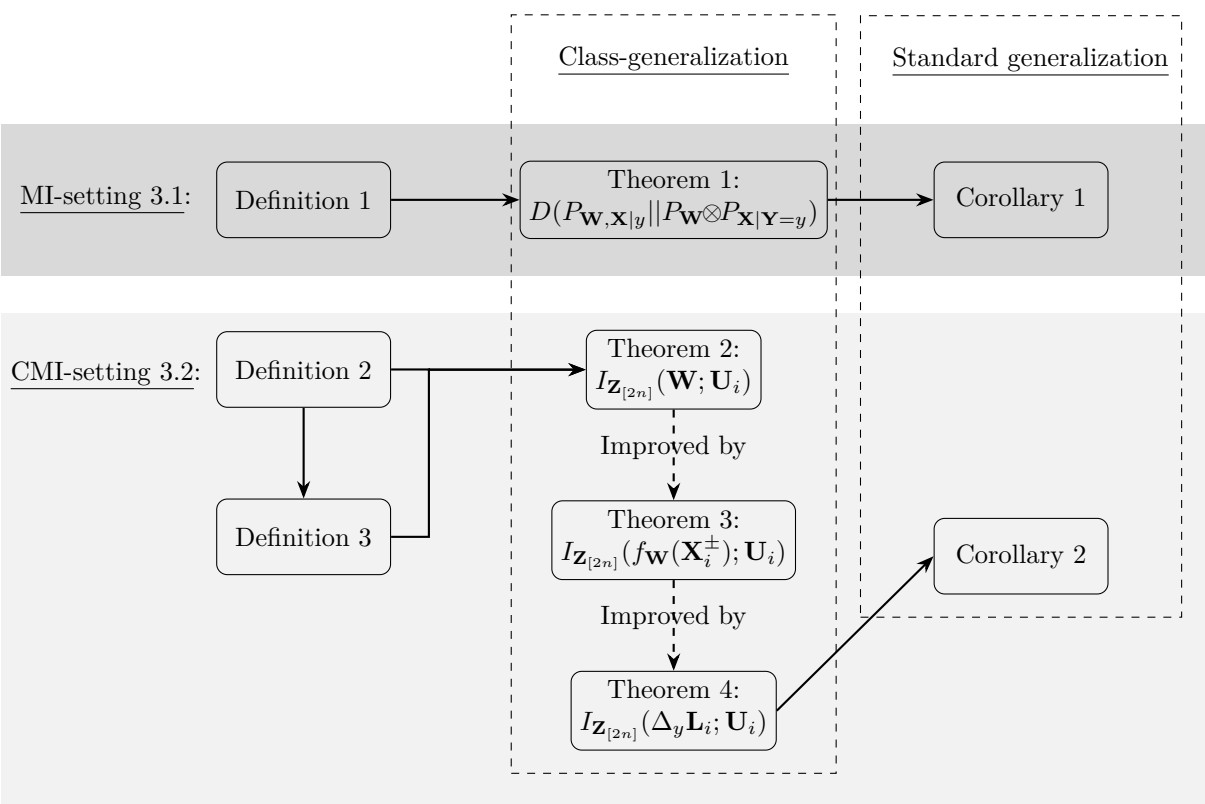

Figure 2: Overview of the main Definitions, Theorems, and Corollaries of Sections 3 and 5.1, in our paper. We use dark gray to refer the MI setting and the light gray to refer the supersample setting

In the supervised learning context, several standard generalization error bounds have been proposed based on different information measures, e.g., KL divergence (Zhou et al., 2023), Wasserstein distance (Rodríguez Gálvez et al., 2021), mutual information between the samples and the weights (Xu & Raginsky, 2017; Bu et al., 2020). Recently, it was shown that tighter generalization bounds could be obtained based on the conditional mutual information (CMI) in the super-sample setting (Steinke & Zakynthinou, 2020; Zhou et al., 2022). Based on this framework, Harutyunyan et al. (2021) derived $f$-CMI bounds based on the model output. In Hellström & Durisi (2022), tighter bounds have been obtained based on the CMI of the loss function, which is further tightened by Wang & Mao (2023) using the $\Delta L$ CMI.

**Class-dependent analysis:** Incorporating label information in generalization analysis is not entirely new, given previous literature He & Su (2020); Chen et al. (2020); Deng et al. (2021). For example, in He & Su (2020), the questions "When and how does the update of weights of neural networks using induced gradient at an example impact the prediction of another example?" have been extensively studied and it was observed that the impact is significant if the two samples are from the same class. In Balestriero et al. (2022); Kirichenko et al. (2023); Bitterwolf et al. (2022); Lee et al. (2023), it has been showed that while standard data augmentation techniques (Bishop & Nasrabadi, 2006) help improve overall performance, it yields lower performance on minority classes. From a theoretical perspective, (Deng et al., 2021) noticed that in uniform stability context, the sensitivity of neural networks is highly dependent on the label information and thus proposed the concept of "*Locally Elastic Stability*" to derive tighter algorithmic stability generalization bounds. In Tishby et al. (2000), an information bottleneck principle is proposed, which states that an optimal feature map simultaneously minimizes its mutual information with the feature distribution and maximizes its mutual information with the label distribution, thus incorporating the class information. In Tishby & Zaslavsky (2015); Saxe et al. (2019); Kawaguchi et al. (2023), this principle was used to explain the generalization of neural networks. It is also worth mentioning the work of Morvant et al. (2012), which focuses on bounding the entries of the confusion matrix to understand multi-class classification with PAC-

Bayesian bounds. In contrast, our work takes a different perspective. Specifically, we introduce the concept of class-generalization error, which quantifies the generalization of a specific class and our theoretical results are not restricted to the Gibbs algorithm, like Morvant et al. (2012), and are valid for any learning algorithm.

## 3 Class-Generalization Error

### 3.1 MI-setting

Typically, in supervised learning, the training dataset $\mathbf{S} = \{(\mathbf{X}_i, \mathbf{Y}_i)\}_{i=1}^n = \{\mathbf{Z}_i\}_{i=1}^n$ contains $n$ i.i.d. samples $\mathbf{Z}_i \in \mathcal{Z}$ generated from the distribution $P_{\mathbf{Z}}$. Here, we are interested in the performance of a model with weights $w \in \mathcal{W}$ for data coming from a specific class $y \in \mathcal{Y}$. To this end, we define $\mathbf{S}_y$ as the subset of $\mathbf{S}$ composed of samples only in class $y$. For any model $w \in \mathcal{W}$ and fixed training sets $s$ and $s_y$, the class-wise empirical risk can be defined as follows:

$$L_E(w, s_y) = \frac{1}{n_y} \sum_{(x_i, y) \in s_y} \ell(w, x_i, y), \tag{1}$$

where $n_y$ is the size of $s_y$ ($n_y \leq n$), and $\ell : \mathcal{W} \times \mathcal{X} \times \mathcal{Y} \to \mathbb{R}_0^+$ is a non-negative loss function. In addition, the class-wise population risk that quantifies how well $w$ performs on the conditional data distribution $P_{\mathbf{X}|\mathbf{Y}=y}$ is defined as

$$L_P(w, P_{\mathbf{X}|\mathbf{Y}=y}) = \mathbb{E}_{P_{\mathbf{X}|\mathbf{Y}=y}}[\ell(w, \mathbf{X}, y)]. \tag{2}$$

A learning algorithm can be characterized by a randomized mapping from the entire training dataset $\mathbf{S}$ to model weights $\mathbf{W}$ according to a conditional distribution $P_{\mathbf{W}|\mathbf{S}}$. The gap between $L_P(w, P_{\mathbf{X}|\mathbf{Y}=y})$ and $L_E(w, s_y)$ measures how well the trained model $\mathbf{W}$ overfits the training data with label $y$, and the expected class-generalization error is formally defined as follows.

**Definition 1.** *(class-generalization error) Given $y \in \mathcal{Y}$, the class-generalization error is*

$$\overline{\mathrm{gen}}_y(P_{\mathbf{X},\mathbf{Y}}, P_{\mathbf{W}|\mathbf{S}}) \triangleq \mathbb{E}_{P_{\mathbf{W}}}[L_P(\mathbf{W}, P_{\mathbf{X}|\mathbf{Y}=y})] - \mathbb{E}_{P_{\mathbf{W},S_y}}[L_E(\mathbf{W}, \mathbf{S}_y)], \tag{3}$$

*where $P_{\mathbf{W}}$ and $P_{\mathbf{W},S_y}$ are marginal distributions induced by the learning algorithm $P_{\mathbf{W}|\mathbf{S}}$ and data generating distribution $P_{\mathbf{S}}$.*

**KL divergence bound** For most learning algorithms used in practice, e.g., Stochastic Gradient descent (SGD), the index of training samples $i$ will not affect the distribution of the learned model due to the random batch selection. Thus, similar to prior works (Bu et al., 2020; Zhou et al., 2022), we assume that the conditional distribution $P_{\mathbf{W}|\mathbf{Z}_i}$ obtained from the entire learning algorithm $P_{\mathbf{W}|\mathbf{S}}$, satisfies $P_{\mathbf{W}|\mathbf{Z}_i} = P_{\mathbf{W}|\mathbf{Z}_j}$, $\forall i \neq j$, when each $\mathbf{Z}_i$ is i.i.d. drawn from $P_{\mathbf{Z}}$. Under this assumption, we have the following lemma that simplifies the class-generalization error

**Lemma 1.** *The class-generalization error in definition 1 is given by*

$$\overline{\mathrm{gen}}_y(P_{\mathbf{X},\mathbf{Y}}, P_{\mathbf{W}|\mathbf{S}}) = \mathbb{E}_{P_{\overline{\mathbf{W}}} \otimes P_{\overline{\mathbf{X}}|y}}[\ell(\overline{\mathbf{W}}, \overline{\mathbf{X}}, y)] - \mathbb{E}_{P_{\mathbf{W},\mathbf{X}|y}}[\ell(\mathbf{W}, \mathbf{X}, y)], \tag{4}$$

*where $P_{\mathbf{W},\mathbf{X}|y}$ is the conditional distribution of the shared $P_{\mathbf{W},\mathbf{Z}}$ given $\mathbf{Y} = y$.*

The proof is available in Appendix A.1. Lemma 1 shows that, similar to the standard generalization error (Xu & Raginsky, 2017; Bu et al., 2020; Zhou et al., 2022), the class-wise generalization error can be expressed as the difference between the loss evaluated under the joint distribution and the product-of-marginal distribution. The key difference is that both expectations are taken with respect to conditional distributions given $\mathbf{Y} = y$.

The following Theorem provides an upper bound for the class-generalization error in Definition 1.

**Theorem 1.** *For $y \in \mathcal{Y}$, assume the loss $\ell(\overline{\mathbf{W}}, \overline{\mathbf{X}}, y)$ is $\sigma_y$ sub-Gaussian under $P_{\overline{\mathbf{W}}} \otimes P_{\overline{\mathbf{X}}|y}$, then the class-generalization error of class $y$ in Definition 1 can be bounded as:*

$$|\overline{\mathrm{gen}}_y(P_{\mathbf{X},\mathbf{Y}}, P_{\mathbf{W}|\mathbf{S}})| \leq \sqrt{2\sigma_y^2 D(P_{\mathbf{W},\mathbf{X}|y} || P_{\mathbf{W}} \otimes P_{\mathbf{X}|\mathbf{Y}=y})}. \tag{5}$$

The full proof is given in Appendix A.2, which utilizes Lemma 1 and Donsker-Varadhan's variational representation of the KL divergence. Theorem 1 shows that the class-generalization error can be bounded using a class-dependent conditional KL divergence. This sheds some light on the puzzling behavior of class-generalization performance, implying that classes with a lower conditional KL divergence between the conditional joint distribution and the product of the marginal distributions tend to generalize better. To our best knowledge, the bound in Theorem 1 is the first label-dependent bound that aims to explain the variation of generalization errors among the different classes.

We note that our bound is obtained by considering the class-generalization gap of each individual sample with label $y$. This approach, as shown in Bu et al. (2020); Zhou et al. (2022); Harutyunyan et al. (2021), yields tighter bounds using the mutual information between an individual sample and the output of the learning algorithm, compared to the conventional bounds relying on the MI of the entire training set and the algorithm's output (Xu & Raginsky, 2017).

## 3.2 Supersample-setting

One limitation of the proposed bound in Theorem 1 is that it can be vacuous and intractable to estimate in practice, as the bound involves a high dimensional entity the model weights $\mathbf{W}$. To this end, the conditional mutual information (CMI) framework, as pioneered by Steinke & Zakynthinou (2020), has been shown in recent studies (Zhou et al., 2022; Wang & Mao, 2023) to offer tighter bounds on generalization error that are always finite even if the $\mathbf{W}$ is high dimensional and continuous.

In this section, we extend our class-wise analysis using the CMI framework. In particular, we assume that there are $n$ super-samples [2] $\mathbf{Z}_{[2n]} = (\mathbf{Z}_1^{\pm}, \cdots, \mathbf{Z}_n^{\pm}) \in \mathcal{Z}^{2n}$ i.i.d generated from $P_{\mathbf{Z}}$. The training data $\mathbf{S} = (\mathbf{Z}_1^{\mathbf{U}_1}, \mathbf{Z}_2^{\mathbf{U}_2}, \cdots, \mathbf{Z}_n^{\mathbf{U}_n})$ are selected from $\mathbf{Z}_{[2n]}$, where $\mathbf{U} = (\mathbf{U}_1, \cdots, \mathbf{U}_n) \in \{-1, 1\}^n$ is the selection vector composed of $n$ independent Rademacher random variables. Intuitively, $\mathbf{U}_i$ selects sample $\mathbf{Z}_i^{\mathbf{U}_i}$ from $\mathbf{Z}_i^{\pm}$ to be used in training, and the remaining one $\mathbf{Z}_i^{-\mathbf{U}_i}$ is for the test.

One potential approach to define class-generalization in the supersample setting is to construct it equivalently to the class generalization error in the MI setting (Definition 1).

**Definition 2.** *(class-generalization error with global $\frac{1}{n^y}$ ) For any $y \in \mathcal{Y}$, the class-generalization error is defined as*

$$\widetilde{\mathrm{gen}}_y \triangleq \frac{1}{n^y} \mathbb{E}_{\mathbf{Z}_{[2n]}} \Big[ \sum_{i=1}^n \mathbb{E}_{\mathbf{U}_i, \mathbf{W} | \mathbf{Z}_{[2n]}} \big[ \mathbb{1}_{\{Y_i^{-U_i} = y\}} \ell(\mathbf{W}, \mathbf{Z}_i^{-\mathbf{U}_i}) - \mathbb{1}_{\{Y_i^{U_i} = y\}} \ell(\mathbf{W}, \mathbf{Z}_i^{\mathbf{U}_i}) \big] \Big], \tag{6}$$

*where $n^y = n P(y)$, $P(y)$ is the true probability of class $y$, and $\mathbb{1}_{\{a=b\}}$ is the indicator function, returning 1 when $a = b$ and zero otherwise.*

We can show the exact equivalence between Definition 1 and Definition 2, with details presented in Appendix B.1. Similarly to Definition 1, the class-generalization error in Definition 2 measures the expected error gap between the training set and the test set relative to one specific class $y$.

As the class-generalization error as defined in Definition 2 depends explicitly on $P(y)$, it has a significant practical limitation: $P(y)$ is not typically available in practice. Consequently, any empirical analysis based on this variant necessitates the estimation of $P(y)$, which in turn introduces an additional layer of estimation bias. To overcome this issue, we propose another variant of class-generalization error within the supersample setting. To this end, given a supersample $z_{[2n]}$ and for a specific class $y \in \mathcal{Y}$, let $n_{z_{[2n]}}^y$ denote *half* the number of samples with class $y$ within $z_{[2n]}$. Using $n_{z_{[2n]}}^y$ instead of $n^y$, class-generalization error becomes:

**Definition 3.** *(super-sample-based class-generalization error) For any $y \in \mathcal{Y}$, the class-generalization error is defined as*

$$\overline{\mathrm{gen}}_y \triangleq \mathbb{E}_{\mathbf{Z}_{[2n]}} \Big[ \frac{1}{n_{\mathbf{Z}_{[2n]}}^y} \sum_{i=1}^n \mathbb{E}_{\mathbf{U}_i, \mathbf{W} | \mathbf{Z}_{[2n]}} \big[ \mathbb{1}_{\{Y_i^{-U_i} = y\}} \ell(\mathbf{W}, \mathbf{Z}_i^{-\mathbf{U}_i}) - \mathbb{1}_{\{Y_i^{U_i} = y\}} \ell(\mathbf{W}, \mathbf{Z}_i^{\mathbf{U}_i}) \big] \Big]. \tag{7}$$

---

[2] In Steinke & Zakynthinou (2020), the term supersample refers to the $\mathbf{Z}_{[2n]}$. Here, it refers to a pair $\mathbf{Z}_i^{\pm}$.

Compared to Definition 2, Definition 3 does not have a direct connection to Definition 1. Its main advantage, however, lies in using $n^y_{\mathbf{Z}_{[2n]}}$, which can be computed for every $Z_{[2n]}$, making it more practical for studying class generalization. Hence, in the rest of this Section and Section 4, we focus mainly on this definition of class-generalization error. Note that all the bounds derived in this section based on Definition 3 can also be obtained for Definition 2 in a similar manner and are therefore omitted for simplicity.

Compared to the standard generalization error definition typically used in the super-sample setting (Steinke & Zakynthinou, 2020; Zhou et al., 2022), we highlight two key differences in Definition 2 and 3: (i) Our class-wise generalization error involves indicator functions to consider only samples belonging to a specific class $y$; (ii) Our generalization error is normalized by $n^y$ (or $n^y_{\mathbf{Z}_{[2n]}}$) instead of the total number of samples $n$.

The indicators are critical in Definition 2 and 3, serving the vital purpose of delimiting errors relative to the class of interest $y$. It is worth noting that alternative definitions for class generalization, aside from Definitions 3 and 2, also exist: a notion of class generalization error could be defined using a single indicator function by making each pair of super-samples have the same label, i.e., $\mathbf{Y}_i^+ = \mathbf{Y}_i^-$. However, this alternative requires a fundamental modification of the supersample setting and lacks direct insights into the interrelation between class generalization and standard generalization errors. In contrast, Definition 2, as illustrated later in Section 5.1, not only provides a direct connection to the standard generalization error but also enables us to derive the first label-dependent standard generalization bounds. A detailed discussion of the technical concerns for this alternative is provided in Appendix B.2.

The loss term involved in Definition 3, i.e., $\mathbb{1}_{\{Y_i^{-U_i}=y\}}\ell(\mathbf{W}, \mathbf{Z}_i^{-\mathbf{U}_i}) - \mathbb{1}_{\{Y_i^{U_i}=y\}}\ell(\mathbf{W}, \mathbf{Z}_i^{\mathbf{U}_i})$ has a specific dependency with respect to the indicators. Thus, prior techniques (Wang & Mao, 2023; Harutyunyan et al., 2021) designed for any generic loss function yield loose bounds. We provide a novel CMI-based bound by exploring the structure of these indicator functions in the loss function. The main technical result is presented in Lemma 2.

**Lemma 2.** *Consider the super-sample setting, for a fixed $z_{[2n]}$, let $\mathbf{V} \in \mathcal{V}$ be a random variable depending on the learned weights $\mathbf{W}$. For any function $g$ that can be written as $g(\mathbf{V}, \mathbf{U}_i, z_{[2n]}) = \mathbb{1}_{\{y^{\mathbf{U}_i}=y\}}h(\mathbf{V}, z_i^{\mathbf{U}_i}) - \mathbb{1}_{\{y^{-\mathbf{U}_i}=y\}}h(\mathbf{V}, z_i^{-\mathbf{U}_i})$, where $h \in [0,1]$ is a bounded function, we have*

$$\mathbb{E}_{\mathbf{V}, \mathbf{U}_i|\mathbf{Z}_{[2n]}=z_{[2n]}}[g(\mathbf{V}, \mathbf{U}_i, z_{[2n]})] \leq \sqrt{2\max(\mathbb{1}_{\{y_i^-=y\}}, \mathbb{1}_{\{y_i^+=y\}})I_{z_{[2n]}}(\mathbf{V}; \mathbf{U}_i)}. \tag{8}$$

The presence of the indicator functions introduces a notable technical complexity, as they depend on both $\mathbf{U}_i$ and $\mathbf{Y}_i$. The proof is based on Donsker-Varadhan's variational representation of the KL divergence and Hoeffding's Lemma (Hoeffding, 1994) and is provided in Appendix A.3. Notably, Lemma 2 forms the foundational element for all subsequent bounds in Theorems 2 and 3.

**Class-CMI bound.** The following theorem provides a bound for the super-sample-based class-generalization error using the disintegrated conditional mutual information between $\mathbf{W}$ and the selection variable $\mathbf{U}_i$ conditioned on super-sample $\mathbf{Z}_{[2n]}$.

**Theorem 2** (class-CMI). *Assume that the loss $\ell(w, x, y) \in [0,1]$ is bounded, then the class-generalization error for class $y$ in Definition 3 can be bounded as*

$$|\overline{\mathrm{gen}_y}| \leq \mathbb{E}_{\mathbf{Z}_{[2n]}}\left[\frac{1}{n^y_{\mathbf{Z}_{[2n]}}} \sum_{i=1}^n \sqrt{2\max(\mathbb{1}_{\{\mathbf{Y}_i^-=y\}}, \mathbb{1}_{\{\mathbf{Y}_i^+=y\}})I_{\mathbf{Z}_{[2n]}}(\mathbf{W}; \mathbf{U}_i)}\right]. \tag{9}$$

The full proof is provided in Appendix A.4. Theorem 2 provides a bound of the class-generalization error with explicit dependency on the weights $\mathbf{W}$, which implies that the class-generalization error depends on how much information the random selection reveals about the weights when at least one of the two samples of $z_i^{\pm}$ corresponds to the class of interest $y$. Note that the links between overfitting and memorization have been established in Zhang et al. (2016); Arpit et al. (2017); Chatterjee (2018). Here, Theorem 2 highlights the role of model sensitivity to the training data—specifically, a high conditional mutual information between $\mathbf{W}$ and $\mathbf{U}_i$ indicates that changes in the selection of training data lead to significant changes in the learned model, resulting in looser bounds and potentially worse generalization for some classes.

**Class-f-CMI bound.** While the bound in Theorem 2 is always finite as $\mathbf{U}_i$ is binary, evaluating $I_{\mathbf{Z}_{[2n]}}(\mathbf{W}; \mathbf{U}_i)$ can be challenging, especially when $\mathbf{W}$ is high-dimensional as in deep networks. One way to overcome this issue is by considering the predictions of the model $f_{\mathbf{W}}(\mathbf{X}_i^{\pm})$ instead of the model weights $\mathbf{W}$, as proposed by Harutyunyan et al. (2021). Here, we denote the loss function $\ell$ based on the prediction $\hat{y} = f_w(x)$ as $\ell(w, x, y) = \ell(\hat{y}, y) = \ell(f_w(x), y)$. Throughout the rest of the paper, we use these two notations of loss interchangeably when it is clear from the context.

In the following theorem, we bound the class-generalization error based on the disintegrated CMI between the model prediction $f_{\mathbf{W}}(\mathbf{X}_i^{\pm})$ and the random selection, i.e., $I_{\mathbf{Z}_{[2n]}}(f_{\mathbf{W}}(\mathbf{X}_i^{\pm}); \mathbf{U}_i)$.

**Theorem 3.** *(class-f-CMI) Assume that the loss $\ell(\hat{y}, y) \in [0, 1]$ is bounded, then the class-generalization error for class $y$ in Definition 3 can be bounded as*

$$|\overline{\mathrm{gen}_y}| \leq \mathbb{E}_{\mathbf{Z}_{[2n]}} \left[ \frac{1}{n_{\mathbf{Z}_{[2n]}}^y} \sum_{i=1}^{n} \sqrt{2 \max(\mathbb{1}_{\{\mathbf{Y}_i^- = y\}}, \mathbb{1}_{\{\mathbf{Y}_i^+ = y\}}) I_{\mathbf{Z}_{[2n]}}(f_{\mathbf{W}}(\mathbf{X}_i^{\pm}); \mathbf{U}_i)} \right]. \tag{10}$$

*Moreover, the class-f-CMI bound is always tighter than the class-CMI bound in Theorem 2.*

The proof is available in Appendix A.5. The main benefit of the class-f-CMI bound, compared to all previously presented bounds, lies in the evaluation of the CMI term involving a low-dimensional random variable $f_{\mathbf{W}}(\mathbf{X}_i^{\pm})$ and a binary random variable $\mathbf{U}_i$. For example, in the case of binary classification, $f_{\mathbf{W}}(\mathbf{X}_i^{\pm})$ will be a pair of two binary variables, which enables us to estimate the class-f-CMI bound efficiently and accurately, as will be shown in Section 4.

**Remark 1.** *In contrast to the bound in Theorem 2, the bound in Theorem 3 does not require access to the model parameters $\mathbf{W}$. It only requires the model output $f(\cdot)$, which makes it suitable even for non-parametric approaches or black-box evaluation.*

**Remark 2.** *An issue of both bounds in Theorems 2 and 3 is that they depend on information quantities irrelevant to the class $y$. The term $\max(\mathbb{1}_{\{\mathbf{Y}_i^- = y\}}, \mathbb{1}_{\{\mathbf{Y}_i^+ = y\}})$ filters out the CMI terms where neither sample $\mathbf{Z}_i^+$ nor $\mathbf{Z}_i^-$ corresponds to the class $y$. However, this term does not require both samples $\mathbf{Z}_i^{\pm}$ to belong to class $y$. In the case that one sample in the pair $(\mathbf{Z}_i^-, \mathbf{Z}_i^+)$ is from class $y$ and the other is from a different class, this term is non-zero and the information from both samples of the pair, i.e., $I_{\mathbf{Z}_{[2n]}}(f_{\mathbf{W}}(\mathbf{X}_i^{\pm}); \mathbf{U}_i)$, contributes to the bound. From this perspective, samples from other classes ($\neq y$) can still affect the bounds, potentially leading to less tight bounds for class $y$.*

**Class-$\Delta_y L$-CMI bound.** In the following, we show that it is possible to address the issue discussed in Remark 2. To this end, we consider a new random variable $\Delta_y \mathbf{L}_i$ based on the indicator function and the loss, which is defined as $\Delta_y \mathbf{L}_i \triangleq \mathbb{1}_{\{y_i^- = y\}} \ell(f_{\mathbf{W}}(\mathbf{X}_i)^-, y_i^-) - \mathbb{1}_{\{y_i^+ = y\}} \ell(f_{\mathbf{W}}(\mathbf{X}_i)^+, y_i^+)$. As shown in Wang & Mao (2023); Hellström & Durisi (2022), using the difference of the loss functions on $\mathbf{Z}_i^{\pm}$ instead of the model output yields tighter generalization bounds for the standard generalization error. In addition, this $\Delta_y \mathbf{L}_i$ only subsumes terms related to class $y$, which further tightens the bound for class-wise generalization. The following Theorem provides a bound based on the CMI using the newly introduced variable.

**Theorem 4.** *(class-$\Delta_y L$-CMI) Assume that the loss $\ell(\hat{y}, y) \in [0, 1]$ is bounded, then the class-generalization error of class $y$ defined in 3 can be bounded as*

$$|\overline{\mathrm{gen}_y}| \leq \mathbb{E}_{\mathbf{Z}_{[2n]}} \left[ \frac{1}{n_{\mathbf{Z}_{[2n]}}^y} \sum_{i=1}^{n} \sqrt{2 I_{\mathbf{Z}_{[2n]}}(\Delta_y \mathbf{L}_i; \mathbf{U}_i)} \right]. \tag{11}$$

*Moreover, the $\Delta_y L$-CMI bound is always tighter than the class-f-CMI bound in Theorem 3.*

The proof is available in Appendix A.7. Unlike Theorem 3, the bound in Theorem 4 does not directly rely on the model output $f(\cdot)$. Instead, it only requires the loss values for $\mathbf{Z}_i^{\pm}$ to compute $\Delta_y \mathbf{L}_i$. Intuitively, the difference between two weighted loss values, $\Delta_y \mathbf{L}_i$, reveals much less information about the selection process $\mathbf{U}_i$ compared to the pair $f_{\mathbf{W}}(\mathbf{X}_i^{\pm})$. In Appendix A.7, we formally show that indeed the $\Delta_y L$-CMI bound is always tighter than the class-f-CMI bound. Another key advantage of Theorem 4 is that computing the

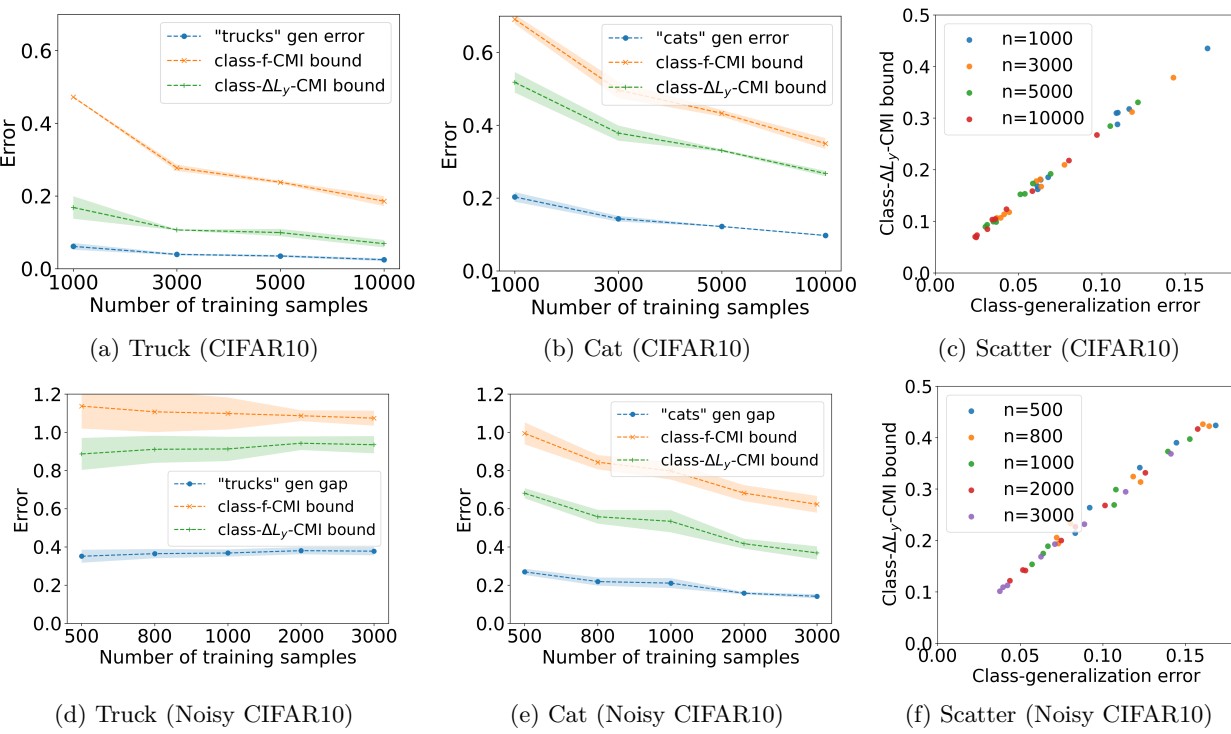

Figure 3: Experimental results of class-generalization error and our bounds in Theorems 3 and 4 for CIFAR10 and noisy CIFAR10, as we increase the number of training samples $n$. The right column shows the relationship between the bound in Theorem 4 and the true class-generalization error for all classes in each dataset.

CMI term $I_{\mathbf{Z}_{[2n]}}(\Delta_y \mathbf{L}_i; \mathbf{U}_i)$ is even simpler, given that $\Delta_y \mathbf{L}_i$ is a one-dimensional scalar, as opposed to the two-dimensional $f_{\mathbf{W}}(\mathbf{X}_i^{\pm})$.

Interestingly, it should also be noted that a similar class-$\Delta_y L$-CMI bound can be derived based on the alternative Definition 2 using Theorem 3.1 in Wang & Mao (2023): As their result is valid for any loss, consider in particular the loss $\mathbb{1}_y \ell$. However, note that such proof technique can not be used to derive the Theorem 4 for Definition 3 due to the presence of $n_{\mathbf{Z}_{[2n]}}^y$.

In corroboration with the results in Figure 1, Theorems 2, 3, and 4 show that having more samples from class $y$ (larger $n_y$) cannot guarantee strong class-generalization, as $n_y$ is not the sole factor. Indeed, our bounds highlight a fundamental dependency of the class-generalization error with the CMI between the model and the class data. Moreover, this shows that having a more balanced dataset (equal $n_y$) does not guarantee equal class-generalization error, as it does not guarantee equal relative CMI, which can explain the observed disparity of overfitting as shown in Figure 1 and Balestriero et al. (2022); Kirichenko et al. (2023).

## 4 Empirical Evaluations

In this section, we empirically evaluate the effectiveness of our class-wise generalization error bounds. As mentioned earlier, The bounds in Section 3.2 are significantly easy to estimate in practical scenarios. Here, we evaluate the error bounds in Theorems 3 and 4 for deep neural networks. We follow the same experimental settings in Harutyunyan et al. (2021), i.e., we fine-tune a ResNet-50 (He et al., 2016) on the CIFAR10 dataset (Krizhevsky et al., 2009) (pretrained (Schmidhuber, 1992) on ImageNet (Deng et al., 2009)). Moreover, to understand how well our bounds perform in a more challenging situation and to further highlight their effectiveness, we conduct an additional experiment with a noisy variant (5% label noise) of CIFAR10. The details are provided in Appendix C.1.

The class-wise generalization error of two classes from CIFAR10 "trucks" and "cats", along with the bounds in Theorems 3 and 4 are presented in the first two columns of Figure 3. The results on all the ten classes for both datasets, along with additional experiments, are presented in Appendix C.3. Figure 3 shows that both bounds can capture the behavior of the class-generalization error. As expected, the class-$\Delta_y L$-CMI is consistently tighter and more stable compared to the class-$f$-CMI bound for all the different scenarios. For CIFAR10 in Figure 3 (top), as we increase the number of training samples, the "trucks" class has a relatively constant class-generalization error, while the "cats" class has a large slope at the start and then a steady incremental decrease. For both classes, the class-$\Delta_y L$-CMI precisely captures the behavior of class-generalization error.

The results on noisy CIFAR10 in Figure 3 (bottom) and the results in Appendix C.3 are consistent with these observations. Notably, the "trucks" generalization error decreases for CIFAR10 and increases for noisy CIFAR10 with respect to the number of samples. Moreover, the class-generalization error of "cat" is worse than "trucks" in CIFAR10, but the opposite is true for the noisy CIFAR10. Our class-$\Delta_y L$-CMI bound successfully captures all these complex behaviors.

The left and middle plots in Figure 3 show that the class-$\Delta_y L$-CMI bound scales proportionally with the actual class-generalization error, i.e., higher class-$\Delta_y L$-CMI bound value indicate a higher class-generalization error. To further highlight this dependency, Figure 3 (right) presents the scatter plot between the different class-generalization errors and their corresponding class-$\Delta_y L$ bound values for all classes in CIFAR10 (top) and Noisy CIFAR10 (bottom) under different number of samples. Our bound is linearly correlated with the true error and can efficiently predict its behavior. A similar pattern is observed for the $f$-CMI bound, as detailed in Appendix C.3. Further validation on the more complex CIFAR100 dataset, provided in Appendix C.4, confirms the bounds' capacity to effectively capture class-specific generalization patterns.

**Class-generalization in traditional ML approaches:** Although the primary focus of this paper is class-generalization in neural networks, it is worth noting that our theoretical results are valid for any random learning algorithm, including classic approaches such as SVM and decision trees. The empirical results of these two models with MNIST are available in Appendix C.5. The empirical results further corroborate our findings and show that our bounds are generic and effectively capture the class-generalization behavior of traditional ML algorithms. These results highlight that class-wise disparities exist even in low-capacity models, and our bounds can meaningfully quantify them.

**Recall and Specificity:** Standard generalization bounds (Xu & Raginsky, 2017; Harutyunyan et al., 2021) focus on classification error or accuracy, but these metrics are often inadequate for imbalanced datasets. In detecting rare cancers, for example, recall and specificity are more relevant performance measures. However, existing bounds (Wu et al., 2020; Wang & Mao, 2023) offer no theoretical insights into these metrics. This paper addresses this gap by providing a framework to analyze generalization in terms of recall and specificity. In the special case of binary classification with 0-1 loss, the class-generalization errors studied here correspond to recall and specificity, as detailed in Appendix C.6. Empirical results on MNIST, presented in Figure 4, validate the tightness of our bounds and their utility in providing generalization certificates for recall and specificity, essential in sensitive applications. This connection enables principled evaluation of critical metrics in domains where traditional accuracy-based bounds are insufficient.

To sum up, (i) one can use our bound to predict which classes will generalize better than others or which classes can benifit from having more data; (ii) in corroboration with theoretical results in Section 3, the MI/CMI between the model and the class data can be used as a proxy for class-generalization error. Such a result provides a new perspective on improving class-generalization by reducing MI/CMI. The initial empirical results in Appendix C.8 show that this is a promising research direction to mitigate the class-generalization disparity.

## 5 Other Applications

Besides enabling us to study class-wise generalization errors, the tools developed in this work can also be used to provide theoretical insights into several other applications. In this section, we explore several use cases with detailed proofs provided in Appendix D.

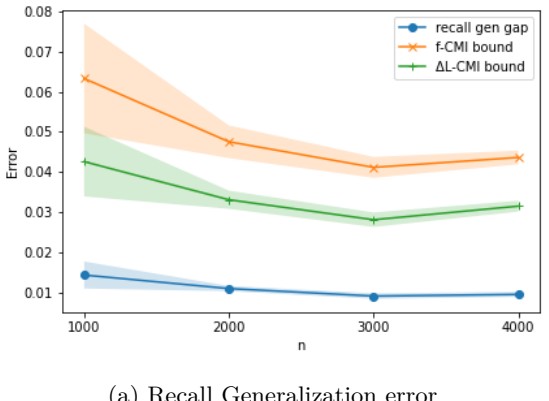 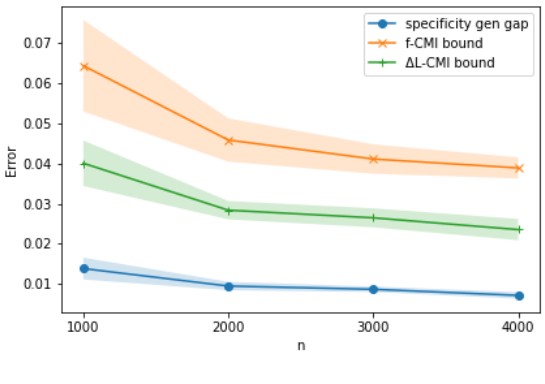

(a) Recall Generalization error

(b) Specificity Generalization error

Figure 4: Recall generalization error (left) and specificity generalization error (right) for the binary classification 4 vs 9 from MNIST. The digit 4 is considered the positive class.

### 5.1 From Class-generalization to Standard Generalization Error

Here, we study the connection between the standard generalization and the class-generalization errors. We extend the bounds presented in Section 3 into class-dependent expected generalization error bounds. First, we notice that taking the expectation over $P_{\mathbf{Y}}$ for the class-generalization error in Definition 1 yields the standard expected generalization error. Thus, we can obtain a class-dependent bound for the standard generalization error by taking the expectation of $y \sim P_{\mathbf{Y}}$ in Theorem 1.

**Corollary 1.** *Assume that for every $y \in \mathcal{Y}$, the loss $\ell(\overline{\mathbf{W}}, \overline{\mathbf{X}}, y)$ is $\sigma_y$ sub-Gaussian under $P_{\overline{\mathbf{W}}} \otimes P_{\overline{\mathbf{X}}|\overline{\mathbf{Y}}=y}$, then*

$$|\overline{\text{gen}}| \leq \mathbb{E}_{\mathbf{Y}'}\left[\sqrt{2\sigma_{\mathbf{Y}'}^2 D(P_{\mathbf{W}|\mathbf{Z}} \otimes P_{\mathbf{X}|\mathbf{Y}=\mathbf{Y}'} || P_{\mathbf{W}} \otimes P_{\mathbf{X}|\mathbf{Y}=\mathbf{Y}'})}\right]. \tag{12}$$

We note that if the sub-Gaussian parameter $\sigma_y = \sigma$ is independent of $y$, we can further show that the bound in 1 is tighter than the individual sample bound in Bu et al. (2020). The proof is provided in Appendix D.1. This shows that the technique proposed in this paper, i.e., deriving bounds by conditioning on the class then converting the bound to standard generalization, can indeed tighten existing information-theoretic bounds in the MI setting. For the supersamples setting, we can use the class-generalization error as Defined in 2, i.e., $\widetilde{\text{gen}}_y$, as for this variant, we have $\overline{\text{gen}} = \mathbb{E}_{\mathbf{Y}}[\widetilde{\text{gen}}_y]$ and hence we can derive the following bound:

**Corollary 2.** *Assume that the loss $\ell(\hat{y}, y) \in [0, 1]$, then*

$$|\overline{\text{gen}}| \leq \mathbb{E}_{\mathbf{Y}}\left[\mathbb{E}_{\mathbf{Z}_{[2n]}}\left[\frac{1}{n^{\mathbf{Y}}}\sum_{i=1}^{n}\sqrt{2I_{\mathbf{Z}_{[2n]}}(\mathbf{\Delta}_{\mathbf{Y}}\mathbf{L}_i; \mathbf{U}_i)}\right]\right].$$

To the best of our knowledge, Corollaries 1 and 2 are the first generalization bounds to provide explicit label-dependency. Although prior bounds (Harutyunyan et al., 2021; Wang & Mao, 2022; 2023) might be tighter or more efficient to estimate, they do not provide any information on how different classes affect the standard generalization error. The results presented here address this gap and provide explicit label-dependent bounds. In $m$-way classification tasks, the bounds become a sum of each class-generalization error weighted by the probability of the class, i.e., $P(\mathbf{Y} = y)$, suggesting that classes with a higher occurrence probability affect the generalization error more. From this perspective, our results can also provide insights into developing algorithms with better generalization by focusing on the class-generalization error. For example, one can employ data augmentation targeted at the classes with higher class-generalization error to attenuate their respective error and thus improve the standard generalization of the model.

## 5.2 Sub-task Problem

*Subtask problem* refers to a specific case of distribution shift in supervised learning, where the training data generated from the source domain $P_{\mathbf{X},\mathbf{Y}}$ consists of multiple classes, while the test data for the target domain $Q_{\mathbf{X},\mathbf{Y}}$ only encompasses a specific known subset of the classes encountered during training. This problem is motivated by the situation where a large model has been trained on numerous classes, potentially over thousands, but is being utilized in a target environment where only a few classes, observed during training, exist. By tackling the problem as a standard domain adaptation task, the generalization error of the subtask problem can be bounded as follows:

$$\overline{\text{gen}}_{Q,E_P} \triangleq \mathbb{E}_{P_{\mathbf{W},\mathbf{s}}}[L_Q(\mathbf{W}) - L_E(\mathbf{W},\mathbf{S})] \leq \sqrt{2\sigma^2 D(Q_{\mathbf{X},\mathbf{Y}}\|P_{\mathbf{X},\mathbf{Y}})} + \sqrt{2\sigma^2 I(\mathbf{W};\mathbf{S})}, \tag{13}$$

where $L_Q(w) = L_P(w, Q_{\mathbf{X},\mathbf{Y}})$ denotes the population risk of $w$ under distribution $Q_{\mathbf{X},\mathbf{Y}}$. We note that (Wu et al., 2020) further tightens the result in equation 13, but these bounds are all based on the KL divergence $D(Q_{\mathbf{X},\mathbf{Y}}\|P_{\mathbf{X},\mathbf{Y}})$ for any generic distribution shift problem and do not leverage the fact that the target task is encapsulated in the source task.

Obtaining tighter generalization error bounds for the subtask problem is straightforward using our class-wise generalization tools. In fact, the generalization error of the subtask can be bounded by summing the class-wise generalization over the space of the subtask classes $\mathcal{A}$. Formally, by taking the expectation of $\mathbf{Y} \sim Q_{\mathbf{Y}}$, we obtain the following notion of the subtask generalization error:

$$\overline{\text{gen}}_{Q,E_Q} \triangleq \mathbb{E}_{Q_{\mathbf{Y}}}[\tilde{\text{gen}}_{\mathbf{Y}}] = \mathbb{E}_{P_{\mathbf{W},\mathbf{s}}}[L_Q(w) - L_{E_Q}(\mathbf{W},\mathbf{S})], \tag{14}$$

where $L_{E_Q}(w,S) = \frac{1}{n_{\mathcal{A}}}\sum_{y_i \in \mathcal{A}} \ell(w, x_i, y_i)$ is the empirical risk relative to the target domain $Q$, and $n_{\mathcal{A}}$ is the number of samples in $S$ such that their labels $y_i \in \mathcal{A}$. We are interested in deriving generalization bounds for $\overline{\text{gen}}_{Q,E_Q}$, as it only differs from $\overline{\text{gen}}_{Q,E_P}$ by the difference in the empirical risk $L_{E_Q}(\mathbf{W},\mathbf{S}) - L_E(\mathbf{W},\mathbf{S})$, which can be computed easily in practice.

As shown in Appendix D.2, we can use the results from Section 3 to obtain tighter bounds. For example, using Theorem 4, we can obtain the subtask generalization error bound in Theorem 5.

**Theorem 5.** *(subtask-$\Delta_y L$-CMI) Assume that the loss $\ell(w, x, y) \in [0, 1]$ is bounded, Then the subtask generalization error defined in equation 14 can be bounded as*

$$|\overline{\text{gen}}_{Q,E_Q}| \leq \mathbb{E}_{\mathbf{Y} \sim Q_{\mathbf{Y}}}\left[\mathbb{E}_{\mathbf{Z}_{[2n]}}\left[\frac{1}{n^{\mathbf{Y}}}\sum_{i=1}^{n}\sqrt{2I_{\mathbf{Z}_{[2n]}}(\Delta_{\mathbf{Y}}\mathbf{L}_i; \mathbf{U}_i)}\right]\right]. \tag{15}$$

Similarly, we can extend Theorems 2 or 3 to construct subtask generalization error bounds using the model's weights or output instead of $\Delta_y \mathbf{L}_i$. In Appendix D.2.1, we empirically validate our subtask bounds and show its ability to capture the generalization behavior in practical subtask scenarios.

**Remark 3.** *Existing distribution shift bounds, e.g., the bound in Eq. 13, typically depend on some measure that quantifies the discrepancy between the true target and true domain distributions, e.g., KL divergence. Note that the difference between Eq. 13 and Eq. 14 is simply the difference between the empirical losses on $E_Q$ and $E_P$. Hence, the bound derived in Theorem 5 can be converted to bounds for Eq. 13 by simply adding $(E_Q - E_P)$ on both sides, which can be directly computed from the training data, eliminating the need for intractable discrepancy measures.*

## 5.3 Generalization Certificates with Sensitive Attributes

One main concern hindering the use of machine learning models in high-stakes applications is the potential biases on sensitive attributes such as gender and skin color (Mehrabi et al., 2021; Barocas et al., 2017). Thus, it is critical not only to reduce the sensitivity to such attributes but also to be able to provide guarantees on the fairness of the models (Holstein et al., 2019; Rajkomar et al., 2018). One aspect of fairness is that the machine learning model should generalize equally well for each group with different sensitive attributes (Barocas et al., 2017; Williamson & Menon, 2019).

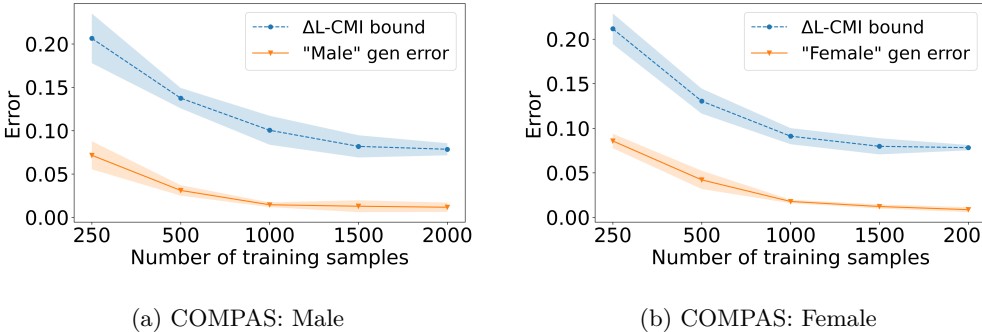

(a) COMPAS: Male          (b) COMPAS: Female

Figure 5: Empirical evaluation of our bounds for generalization with sensitive attributes using using the COMPAS dataset.

By tweaking the definition of our class-generalization error, we show that the theoretical tools developed in this paper can be used to obtain bounds for attribute-generalization errors. Suppose that we have a random variable $\mathbf{T} \in \mathcal{T}$ representing a sensitive feature. One might be interested in studying the generalization of the model for the sub-population with the attribute $\mathbf{T} = t$. Inspired by our class-generalization, we define the attribute-generalization error as follows:

**Definition 4.** *(attribute-generalization error) Given $t \in \mathcal{T}$, the attribute-generalization error is defined as follows:*

$$\overline{\mathrm{gen}_t} = \mathbb{E}_{P_{\mathbf{W}} \otimes P_{\mathbf{Z}|\mathbf{T}=t}}[\ell(\overline{\mathbf{W}}, \overline{\mathbf{Z}})] - \mathbb{E}_{P_{\mathbf{W}|\mathbf{Z}} \otimes P_{\mathbf{Z}|\mathbf{T}=t}}[\ell(\mathbf{W}, \mathbf{Z})]. \tag{16}$$

By exchanging $\mathbf{X}$ and $\mathbf{Y}$ with $\mathbf{Z}$ and $\mathbf{T}$ in Theorem 1, respectively, we can show the following bound for the attribute-generalization error.

**Theorem 6.** *Given $t \in \mathcal{T}$, assume that the loss $\ell(\mathbf{W}, \mathbf{Z})$ is $\sigma$ sub-Gaussian under $P_{\overline{\mathbf{W}}} \otimes P_{\overline{\mathbf{Z}}}$, then the attribute-generalization error of the sub-population $\mathbf{T} = t$, can be bounded as follows:*

$$|\overline{\mathrm{gen}_t}| \leq \sqrt{2\sigma^2 D(P_{\mathbf{W}|\mathbf{Z}} \otimes P_{\mathbf{Z}|t} || P_{\mathbf{W}} \otimes P_{\mathbf{Z}|t})}.$$

We note extending our results to the super-sample settings is also straightforward.

Prior works on fairness in machine learning (Barocas et al., 2019; Han et al., 2024; Giguere et al., 2022) primarily focus on between-group comparisons—evaluating whether a model's performance is consistent across sensitive subpopulations (e.g., male vs. female) and measuring disparities using subpopulation-level metrics such as Equalized Odds (Hardt et al., 2016a) or Demographic Parity (Barocas et al., 2019). In contrast, our theoretical result in Theorem 6 provides a within-subpopulation perspective: it characterizes the generalization behavior of the model on a fixed sensitive group (e.g., conditioned on "male") across the train-test split. This shift in perspective enables a complementary form of fairness analysis. Empirical evaluation of our bound in this case, on **COMPAS** (Larson et al., 2016) dataset are available in Figure 5. Additionally, in Figure 13 of appendix C.7, we provide extra results using **Adult** (Kohavi & Becker, 1996) dataset. As can be seen, our framework yields bounds that consistently capture sub-population based generalization errors, further highlighting its versatility.

Furthermore, using the attribute generalization, we can show that the standard generalization error can be bounded as follows:

**Corollary 3.** *Assume that the loss $\ell(\mathbf{W}, \mathbf{Z})$ is $\sigma$ sub-Gaussian under $P_{\overline{\mathbf{W}}} \otimes P_{\overline{\mathbf{Z}}}$, then*

$$|\overline{\mathrm{gen}}| \leq \mathbb{E}_{\mathbf{T}'} \left[ \sqrt{2\sigma^2 D(P_{\mathbf{W}|\mathbf{Z}} \otimes P_{\mathbf{Z}|\mathbf{T}=\mathbf{T}'} || P_{\mathbf{W}} \otimes P_{\mathbf{Z}|\mathbf{T}=\mathbf{T}'})} \right].$$

The result of Corollary 3 shows that the average generalization error is upper-bounded by the expectation over the attribute-wise generalization. This shows that it is possible to improve the overall generalization by reducing the generalization of each population relative to the sensitive attribute.

# 6 Conclusion & Future Work

This paper studied the puzzle of noticeable disparity of generalization behavior among different classes by introducing and exploring the concept of "class-generalization error". To our knowledge, we provided the first rigorous generalization bounds for this concept using either MI or CMI. We also empirically strengthened the findings with supporting experiments validating the efficiency of the proposed bounds. Furthermore, we demonstrated the versatility of our theoretical tools in providing tight bounds for various contexts.

Overall, our goal is to understand generalization in deep learning through the lens of information theory, which motivates future work on preventing high class-generalization error variability and ensuring 'equal' generalization among the different classes. Other possible future research endeavors focus on obtaining tighter bounds for the class-generalization error, e.g., using the chaining technique (Clerico et al., 2022), and studying this concept in different contexts beyond supervised learning, e.g., self-supervised learning.

## Acknowledgment

The research reported in this publication was supported by funding from King Abdullah University of Science and Technology (KAUST) - Center of Excellence for Generative AI, under award number 5940. This research was also supported by Business Finland (BF)-AMALIA and NFS IUCRC CBL. Yuheng Bu was supported by the Department of Electrical and Computer Engineering at the University of Florida.

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

# A    Proofs of the Theorems in Section 3

This section includes all the proofs of the results presented in the main text in Section 3. We start by the formal definition of sub-Gaussian random variable and the Hoeffding's lemma (Hoeffding, 1994).

**Definition 5.** *A random variable* $\mathbf{X}$ *is called sub-Gaussian if there exists a positive constant* $\sigma > 0$ *such that*

$$\mathbb{E}\left[\exp\left(t(\mathbf{X} - \mathbb{E}[\mathbf{X}])\right)\right] \leq \exp\left(\frac{\sigma^2 t^2}{2}\right) \quad \text{for all } t \in \mathbb{R}. \tag{17}$$

**Lemma 3.** *Let* $\mathbf{X}$ *be a bounded random variable, i.e.,* $\mathbf{X} \in [a,b]$ *almost surely. If* $\mathbb{E}[\mathbf{X}] = 0$, *then* $\mathbf{X}$ *is* $(b-a)$*-sub-Gaussian and we have:*

$$\mathbb{E}[e^{\lambda \mathbf{X}}] \leq e^{\frac{\lambda^2(b-a)^2}{8}}, \quad \forall \lambda \in \mathbb{R}. \tag{18}$$

## A.1    Proof of Lemma 1

With the Assumption that $P_{\mathbf{W},\mathbf{Z}_i} = P_{\mathbf{W},\mathbf{Z}_j}$, $\forall i \neq j$, it follows directly that all the terms in the sum in Definition 1 become identical and we obtain Lemma 1, i.e., the individual-sample-based expression of the class-wise generalization.

**Lemma 1**(restate) The class-generalization error in definition 1 is given by

$$\overline{\mathrm{gen}}_y(P_{\mathbf{X},\mathbf{Y}}, P_{\mathbf{W}|\mathbf{S}}) = \mathbb{E}_{P_{\overline{\mathbf{W}}} \otimes P_{\overline{\mathbf{X}}|y}}[\ell(\overline{\mathbf{W}}, \overline{\mathbf{X}}, y)] - \mathbb{E}_{P_{\mathbf{W},\mathbf{X}|y}}[\ell(\mathbf{W}, \mathbf{X}, y)], \tag{19}$$

where $P_{\mathbf{W},\mathbf{X}|y}$ is the conditional distribution of the shared $P_{\mathbf{W},\mathbf{Z}}$ given $\mathbf{Y} = y$.

*Proof.* Starting from Definition 1, the class-generalization error of class y can be rewritten as

$$\overline{\mathrm{gen}}_y(P_{\mathbf{X},\mathbf{Y}}, P_{\mathbf{W}|\mathbf{S}}) = \mathbb{E}_{P_{\mathbf{W}}}[L_P(\mathbf{W}, P_{\mathbf{X}|\mathbf{Y}=y})] - \mathbb{E}_{P_{\mathbf{W},\mathcal{S}_y}}[L_E(\mathbf{W}, \mathbf{S}_y)]$$

$$= \mathbb{E}_{P_{\mathbf{W}}}[\mathbb{E}_{P_{\mathbf{X}|\mathbf{Y}=y}}[\ell(w, \mathbf{X}, y)]] - \mathbb{E}_{P_{\mathbf{W},\mathcal{S}_y}}[\frac{1}{n_y} \sum_{(x_i,y) \in s_y} \ell(w, x_i, y)]$$

$$= \mathbb{E}_{P_{\overline{\mathbf{W}}} \otimes P_{\overline{\mathbf{X}}|y}}[\ell(w, \mathbf{X}, y)] - \frac{1}{n_y} \sum_{(x_i,y) \in s_y} \mathbb{E}_{P_{\mathbf{W},\mathbf{x}_i|y}}[\ell(w, x_i, y)]. \tag{20}$$

Given the assumption that $P_{\mathbf{W},\mathbf{Z}_i} = P_{\mathbf{W},\mathbf{Z}_j}$, $\forall i \neq j$, we have all the terms in the sum in equation 20 are identical. Thus we have

$$\overline{\mathrm{gen}}_y(P_{\mathbf{X},\mathbf{Y}}, P_{\mathbf{W}|\mathbf{S}}) = \mathbb{E}_{P_{\overline{\mathbf{W}}} \otimes P_{\overline{\mathbf{X}}|y}}[\ell(w, \mathbf{X}, y)] - \frac{1}{n_y} \sum_{(x_i,y) \in s_y} \mathbb{E}_{P_{\mathbf{W},\mathbf{X}|y}}[\ell(w, \mathbf{X}, y)]$$

$$= \mathbb{E}_{P_{\overline{\mathbf{W}}} \otimes P_{\overline{\mathbf{X}}|y}}[\ell(\overline{\mathbf{W}}, \overline{\mathbf{X}}, y)] - \mathbb{E}_{P_{\mathbf{W},\mathbf{X}|y}}[\ell(\mathbf{W}, \mathbf{X}, y)], \tag{21}$$

which completes the proof. $\qquad\square$

## A.2    Proof of Theorem 1

**Theorem 1** (restated) For $y \in \mathcal{Y}$, assume the loss $\ell(\overline{\mathbf{W}}, \overline{\mathbf{X}}, y)$ is $\sigma_y$ sub-Gaussian under $P_{\overline{\mathbf{W}}} \otimes P_{\overline{\mathbf{X}}|\overline{\mathbf{Y}}=y}$, then the class-generalization error of class $y$ in Definition 1 can be bounded as:

$$|\overline{\mathrm{gen}}_y(P_{\mathbf{X},\mathbf{Y}}, P_{\mathbf{W}|\mathbf{S}})| \leq \sqrt{2\sigma_y^2 D(P_{\mathbf{W},\mathbf{X}|\mathbf{Y}=y}||P_{\mathbf{W}} \otimes P_{\mathbf{X}|\mathbf{Y}=y})}. \tag{22}$$

*Proof.* From lemma 1, we have

$$\overline{\mathrm{gen}}_y(P_{\mathbf{X},\mathbf{Y}}, P_{\mathbf{W}|\mathbf{S}}) = \mathbb{E}_{P_{\overline{\mathbf{W}}} \otimes P_{\overline{\mathbf{X}}|\overline{\mathbf{Y}}=y}}[\ell(\overline{\mathbf{W}}, \overline{\mathbf{X}}, y)] - \mathbb{E}_{P_{\mathbf{W},\mathbf{X}|\mathbf{Y}=y}}[\ell(\mathbf{W}, \mathbf{X}, y)]. \tag{23}$$

Using the Donsker–Varadhan variational representation of the relative entropy, we have

$$D(P_{\mathbf{W},\mathbf{X}|\mathbf{Y}=y}||P_{\mathbf{W}}\otimes P_{\mathbf{X}|\mathbf{Y}=y}) \geq \mathbb{E}_{P_{\mathbf{W},\mathbf{X}|\mathbf{Y}=y}}[\lambda\ell(\mathbf{W},\mathbf{X},y)] - \log\mathbb{E}_{P_{\overline{\mathbf{W}}}\otimes P_{\overline{\mathbf{X}}|\overline{\mathbf{Y}}=y}}[e^{\lambda\ell(\overline{\mathbf{W}},\overline{\mathbf{X}},y)}], \tag{24}$$

for all $\lambda \in \mathbb{R}$. On the other hand, we have:

$$\begin{aligned}
&\log\mathbb{E}_{P_{\overline{\mathbf{W}}}\otimes P_{\overline{\mathbf{X}}|\overline{\mathbf{Y}}=y}}\left[e^{\lambda\ell(\overline{\mathbf{W}},\overline{\mathbf{X}},y)-\lambda\mathbb{E}[\ell(\overline{\mathbf{W}},\overline{\mathbf{X}},y)]}\right]\\
&= \log\mathbb{E}_{P_{\overline{\mathbf{W}}}\otimes P_{\overline{\mathbf{X}}|\overline{\mathbf{Y}}=y}}\left[e^{\lambda\ell(\overline{\mathbf{W}},\overline{\mathbf{X}},y)}e^{-\lambda\mathbb{E}[\ell(\overline{\mathbf{W}},\overline{\mathbf{X}},y)]})\right]\\
&= \log\mathbb{E}_{P_{\overline{\mathbf{W}}}\otimes P_{\overline{\mathbf{X}}|\overline{\mathbf{Y}}=y}}[e^{\lambda\ell(\overline{\mathbf{W}},\overline{\mathbf{X}},y)}] - \lambda\mathbb{E}_{P_{\overline{\mathbf{W}}}\otimes P_{\overline{\mathbf{X}}|\overline{\mathbf{Y}}=y}}[\ell(\overline{\mathbf{W}},\overline{\mathbf{X}},y)].
\end{aligned}$$

Using the sub-Gaussian assumption, we have

$$\log\mathbb{E}_{P_{\overline{\mathbf{W}}}\otimes P_{\overline{\mathbf{X}}|\overline{\mathbf{Y}}=y}}[e^{\lambda\ell(\overline{\mathbf{W}},\overline{\mathbf{X}},y)}] \leq \lambda\mathbb{E}_{P_{\overline{\mathbf{W}}}\otimes P_{\overline{\mathbf{X}}|\overline{\mathbf{Y}}=y}}(\ell(\overline{\mathbf{W}},\overline{\mathbf{X}},y)) + \frac{\lambda^2\sigma_y^2}{2}. \tag{25}$$

By replacing in equation 24, we have

$$D(P_{\mathbf{W},\mathbf{X}|\mathbf{Y}=y}||P_{\mathbf{W}}\otimes P_{\mathbf{X}|\mathbf{Y}=y}) \geq \lambda\big(\mathbb{E}_{P_{\mathbf{W},\mathbf{X}|\mathbf{Y}=y}}[\ell(\mathbf{W},\mathbf{X},y)] - \mathbb{E}_{P_{\overline{\mathbf{W}}}\otimes P_{\overline{\mathbf{X}}|\overline{\mathbf{Y}}=y}}[\ell(\overline{\mathbf{W}},\overline{\mathbf{X}},y)]\big) - \frac{\lambda^2\sigma_y^2}{2}.$$

Thus,

$$\begin{aligned}
D(P_{\mathbf{W},\mathbf{X}|\mathbf{Y}=y}||P_{\mathbf{W}}\otimes P_{\mathbf{X}|\mathbf{Y}=y}) - \lambda(\mathbb{E}_{P_{\mathbf{W},\mathbf{X}|\mathbf{Y}=y}}[\ell(\mathbf{W},\mathbf{X},y)] - \mathbb{E}_{P_{\overline{\mathbf{W}}}\otimes P_{\overline{\mathbf{X}}|\overline{\mathbf{Y}}=y}}[\ell(\overline{\mathbf{W}},\overline{\mathbf{X}},y)])\\
+ \lambda^2\sigma_y^2 \geq 0, \qquad \forall\lambda \in \mathbb{R}. \tag{26}
\end{aligned}$$

Equation equation 26 is a non-negative parabola with respect to $\lambda$, which implies its discriminant must be non-positive. Thus,

$$|\mathbb{E}_{P_{\mathbf{W},\mathbf{X}|\mathbf{Y}=y}}[\ell(\mathbf{W},\mathbf{X},y)] - \mathbb{E}_{P_{\overline{\mathbf{W}}}\otimes P_{\overline{\mathbf{X}}|\overline{\mathbf{Y}}=y}}[\ell(\overline{\mathbf{W}},\overline{\mathbf{X}},y)]| \leq \sqrt{2\sigma_y^2 D(P_{\mathbf{W},\mathbf{X}|\mathbf{Y}=y}||P_{\mathbf{W}}\otimes P_{\mathbf{X}|\mathbf{Y}=y})},$$

which completes the proof. $\qquad\square$

### A.3 Proof of Lemma 2

We use Lemma 2 as a main tool to prove Theorem 2, 3, and 7 in Section 3.2 in the super-sample setting.

**Lemma 2** (restated) Consider the super-sample setting, and let $\mathbf{V} \in \mathcal{V}$ be a random variable, possibly depending on $\mathbf{W}$. For any function $g$ that can be written as $g(\mathbf{V},\mathbf{U}_i,z_{[2n]}) = \mathbb{1}_{\{y^{\mathbf{U}_i}=y\}}h(\mathbf{V},z_i^{\mathbf{U}_i}) - \mathbb{1}_{\{y^{-\mathbf{U}_i}=y\}}h(\mathbf{V},z_i^{-\mathbf{U}_i})$, where $h \in [0,1]$ is a bounded function, we have

$$\mathbb{E}_{\mathbf{V},\mathbf{U}_i|\mathbf{Z}_{[2n]}=z_{[2n]}}[g(\mathbf{V},\mathbf{U}_i,z_{[2n]})] \leq \sqrt{2\max(\mathbb{1}_{\{y_i^-=y\}},\mathbb{1}_{\{y_i^+=y\}})I_{z_{[2n]}}(\mathbf{V};\mathbf{U}_i)}. \tag{27}$$

*Proof.* Let $(\overline{\mathbf{V}},\overline{U_i})$ be an independent copy of $(\mathbf{V},\mathbf{U}_i)$. The disintegrated mutual information $I_{z_{[2n]}}(\mathbf{V};\mathbf{U}_i)$ is equal to:

$$I_{z_{[2n]}}(\mathbf{U}_i;\mathbf{V}) = D\big(P_{\mathbf{V},\mathbf{U}_i|\mathbf{Z}_{[2n]}=z_{[2n]}}||P_{\mathbf{V}|\mathbf{Z}_{[2n]}=z_{[2n]}}P_{\mathbf{U}_i}\big), \tag{28}$$

Thus, by the Donsker–Varadhan variational representation of KL divergence, $\forall\lambda \in \mathbb{R}$ and for every function $g$, we have

$$I_{z_{[2n]}}(\mathbf{V};\mathbf{U}_i) \geq \lambda\mathbb{E}_{\mathbf{V},\mathbf{U}_i|\mathbf{Z}_{[2n]}=z_{[2n]}}[g(\mathbf{V},\mathbf{U}_i,z_{[2n]})] - \log\mathbb{E}_{\overline{\mathbf{V}},\overline{\mathbf{U}}_i|\mathbf{Z}_{[2n]}=z_{[2n]}}[e^{\lambda g(\overline{\mathbf{V}},\overline{\mathbf{U}}_i,z_{[2n]})}]. \tag{29}$$

Next, let $g(\mathbf{V},\mathbf{U}_i,z_{[2n]}) = \mathbb{1}_{\{y^{\mathbf{U}_i}=y\}}h(\mathbf{V},z_i^{\mathbf{U}_i}) - \mathbb{1}_{\{y^{-\mathbf{U}_i}=y\}}h(\mathbf{V},z_i^{-\mathbf{U}_i})$. It is easy to see that $g(\overline{\mathbf{V}},\overline{\mathbf{U}}_i,z_{[2n]})$ can be rewritten as follows:

$$g(\overline{\mathbf{V}},\overline{\mathbf{U}}_i,z_{[2n]}) = \overline{\mathbf{U}}_i(\mathbb{1}_{\{y_i^-=y\}}h(\overline{\mathbf{V}},z_i^-) - \mathbb{1}_{\{y_i^+=y\}}h(\overline{\mathbf{V}},z_i^+)). \tag{30}$$

Thus, we have

$$\log \mathbb{E}_{\overline{\mathbf{V}}, \overline{\mathbf{U}}_i | \mathbf{Z}_{[2n]} = z_{[2n]}}[e^{\lambda g(\overline{\mathbf{V}}, \overline{\mathbf{U}}_i, z_{[2n]})}] = \log \mathbb{E}_{\overline{\mathbf{V}}, \overline{\mathbf{U}}_i | \mathbf{Z}_{[2n]} = z_{[2n]}}[e^{\lambda \overline{\mathbf{U}}_i (\mathbb{1}_{\{y_i^- = y\}} h(\overline{\mathbf{V}}, z_i^-) - \mathbb{1}_{\{y_i^+ = y\}} h(\overline{\mathbf{V}}, z_i^+))}].$$

Note that $\mathbb{E}_{\overline{\mathbf{U}}_i}[\overline{\mathbf{U}}_i(\mathbb{1}_{\{y_i^- = y\}} h(\overline{\mathbf{V}}, z_i^-) - \mathbb{1}_{\{y_i^+ = y\}} h(\overline{\mathbf{V}}, z_i^+))] = 0$ and $\overline{\mathbf{U}}_i \in \{-1, +1\}$. By Hoeffding's Lemma, we have

$$\log \mathbb{E}_{\overline{\mathbf{V}}, \overline{\mathbf{U}}_i | \mathbf{Z}_{[2n]} = z_{[2n]}}[e^{\lambda g(\overline{\mathbf{V}}, \overline{\mathbf{U}}_i, z_{[2n]})}] \leq \log \mathbb{E}_{\overline{\mathbf{V}} | \mathbf{Z}_{[2n]} = z_{[2n]}}[e^{\frac{\lambda^2}{2}\left(\mathbb{1}_{\{y_i^- = y\}} h(\overline{\mathbf{V}}, z_i^-) - \mathbb{1}_{\{y_i^+ = y\}} h(\overline{\mathbf{V}}, z_i^+)\right)^2}].$$

As $h \in [0, 1]$, $\left|\mathbb{1}_{\{y_i^- = y\}} h(\overline{\mathbf{V}}, z_i^-) - \mathbb{1}_{\{y_i^+ = y\}} h(\overline{\mathbf{V}}, z_i^+)\right| \leq \max(\mathbb{1}_{\{y_i^- = y\}}, \mathbb{1}_{\{y_i^+ = y\}})$. Thus,

$$\log \mathbb{E}_{\overline{\mathbf{V}}, \overline{\mathbf{U}}_i | \mathbf{Z}_{[2n]} = z_{[2n]}}[e^{\lambda g(\overline{\mathbf{V}}, \overline{\mathbf{U}}_i, z_{[2n]})}] \leq \frac{\lambda^2}{2} \max(\mathbb{1}_{\{y_i^- = y\}}, \mathbb{1}_{\{y_i^+ = y\}})^2 = \frac{\lambda^2}{2} \max(\mathbb{1}_{\{y_i^- = y\}}, \mathbb{1}_{\{y_i^+ = y\}}).$$

Replacing in equation 29, we have

$$I_{z_{[2n]}}(\mathbf{V}; \mathbf{U}_i) \geq \lambda \mathbb{E}_{\mathbf{V}, \mathbf{U}_i | \mathbf{Z}_{[2n]} = z_{[2n]}}[\mathbb{1}_{y^{\mathbf{U}_i} = y} h(\mathbf{V}, z_i^{\mathbf{U}_i}) - \mathbb{1}_{\{y^{-\mathbf{U}_i} = y\}} h(\mathbf{W}, z_i^{-\mathbf{U}_i})] \tag{31}$$
$$- \frac{\lambda^2}{2} \max(\mathbb{1}_{\{y_i^- = y\}}, \mathbb{1}_{\{y_i^+ = y\}}).$$

Therefore, $\forall \lambda \in \mathbb{R}$,

$$\frac{\lambda^2}{2} \max(\mathbb{1}_{\{y_i^- = y\}}, \mathbb{1}_{\{y_i^+ = y\}}) - \lambda \mathbb{E}_{\mathbf{V}, \mathbf{U}_i | \mathbf{Z}_{[2n]} = z_{[2n]}}[g(\mathbf{V}, \mathbf{U}_i, z_{[2n]})] + I_{z_{[2n]}}(\mathbf{V}; \mathbf{U}_i) \geq 0. \tag{32}$$

The equation 32 is a non-negative parabola with respect to $\lambda$. Thus, its discriminant must be non-positive, which implies

$$\mathbb{E}_{\mathbf{V}, \mathbf{U}_i | \mathbf{Z}_{[2n]} = z_{[2n]}}[g(\mathbf{V}, \mathbf{U}_i, z_{[2n]})] \leq \sqrt{2 \max(\mathbb{1}_{\{y_i^- = y\}}, \mathbb{1}_{\{y_i^+ = y\}}) I_{z_{[2n]}}(\mathbf{V}; \mathbf{U}_i)}. \tag{33}$$

$\square$

## A.4 Proof of Theorem 2

**Theorem 2** (restated) Assume that the loss $\ell(w, x, y) \in [0, 1]$ is bounded, then the class-generalization error for class $y$ in Definition 3 can be bounded as

$$|\overline{\text{gen}}_y(P_{\mathbf{X}, \mathbf{Y}}, P_{\mathbf{W}|\mathbf{S}})| \leq \mathbb{E}_{\mathbf{Z}_{[2n]}}\left[\frac{1}{n_{\mathbf{Z}_{[2n]}}^y} \sum_{i=1}^n \sqrt{2 \max(\mathbb{1}_{\{\mathbf{Y}_i^- = y\}}, \mathbb{1}_{\{\mathbf{Y}_i^+ = y\}}) I_{\mathbf{Z}_{[2n]}}(\mathbf{W}; \mathbf{U}_i)}\right]. \tag{34}$$

*Proof.* Using Lemma 2 with $\mathbf{V} = \mathbf{W}$ and $h(\mathbf{V}, z) = \ell(\mathbf{W}, z)$ in, we have

$$\mathbb{E}_{\mathbf{W}; \mathbf{U}_i | \mathbf{Z}_{[2n]} = z_{[2n]}}[g(\mathbf{W}, \mathbf{U}_i, z_{[2n]})] \leq \sqrt{2 \max(\mathbb{1}_{\{y_i^- = y\}}, \mathbb{1}_{\{y_i^+ = y\}}) I_{z_{[2n]}}(\mathbf{W}; \mathbf{U}_i)}, \tag{35}$$

where $g(\mathbf{W}, \mathbf{U}_i, z_{[2n]}) = \mathbb{1}_{\{y^{\mathbf{U}_i} = y\}} \ell(\mathbf{W}, z_i^{\mathbf{U}_i}) - \mathbb{1}_{\{y^{-\mathbf{U}_i} = y\}} \ell(\mathbf{W}, z_i^{-\mathbf{U}_i})$. Thus, by summing over the different terms in Definition 3 and taking expectation over $\mathbf{Z}_{[2n]}$,

$$|\overline{\text{gen}}_y(P_{\mathbf{X}, \mathbf{Y}}, P_{\mathbf{W}|\mathbf{S}})| \leq \mathbb{E}_{\mathbf{Z}_{[2n]}}\left[\frac{1}{n_{\mathbf{Z}_{[2n]}}^y} \sum_{i=1}^n \sqrt{2 \max(\mathbb{1}_{\{\mathbf{Y}_i^- = y\}}, \mathbb{1}_{\{\mathbf{Y}_i^+ = y\}}) I_{\mathbf{Z}_{[2n]}}(\mathbf{W}; \mathbf{U}_i)}\right]. \tag{36}$$

$\square$

### A.5 Proof of Theorem 3

**Theorem 3** (restated) Assume that the loss $\ell(\mathbf{W}, \mathbf{X}, y) \in [0, 1]$, then the class-generalization error of class $y$ in Definition 3 can be bounded as

$$|\overline{\text{gen}_y}(P_{\mathbf{X}, \mathbf{Y}}, P_{\mathbf{W}|\mathbf{S}})| \leq \mathbb{E}_{\mathbf{Z}_{[2n]}}\Big[\frac{1}{n_{\mathbf{Z}_{[2n]}}^y} \sum_{i=1}^n \sqrt{2\max(\mathbb{1}_{\{\mathbf{Y}_i^- = y\}}, \mathbb{1}_{\{\mathbf{Y}_i^+ = y\}}) I_{\mathbf{Z}_{[2n]}}(f_{\mathbf{W}}(\mathbf{X}_i^\pm); \mathbf{U}_i)}\Big]. \tag{37}$$

Moreover, the class-$f$-CMI bound is always tighter than the class-CMI bound in Theorem 2.

*Proof.* Similar to the proof of Theorem 2. Using Lemma 2 with $\mathbf{V} = f_{\mathbf{W}}(x_i^\pm)$ and $h(\mathbf{V}, z_i) = \ell(f_{\mathbf{W}}(x_i), y_i)$ in, we have

$$\mathbb{E}_{f_{\mathbf{W}}(x_i^\pm); \mathbf{U}_i|\mathbf{Z}_{[2n]} = z_{[2n]}}[g(f_{\mathbf{W}}(\mathbf{X}_i), \mathbf{U}_i, z_{[2n]})] \leq \sqrt{2\max(\mathbb{1}_{\{y_i^- = y\}}, \mathbb{1}_{\{y_i^+ = y\}}) I_{z_{[2n]}}(f_{\mathbf{W}}(x_i^\pm); \mathbf{U}_i)}. \tag{38}$$

Thus taking expectation with respect to $\mathbf{Z}_{[2n]}$ yields the desired result

$$|\overline{\text{gen}_y}(P_{\mathbf{X}, \mathbf{Y}}, P_{\mathbf{W}|\mathbf{S}})| \leq \mathbb{E}_{\mathbf{Z}_{[2n]}}\Big[\frac{1}{n_{\mathbf{Z}_{[2n]}}^y} \sum_{i=1}^n \sqrt{2\max(\mathbb{1}_{\{\mathbf{Y}_i^- = y\}}, \mathbb{1}_{\{\mathbf{Y}_i^+ = y\}}) I_{\mathbf{Z}_{[2n]}}(f_{\mathbf{W}}(\mathbf{X}_i^\pm); \mathbf{U}_i)}\Big]. \tag{39}$$

Due to the data processing inequality, we have $\mathbf{U} \to \mathbf{W} \to f_{\mathbf{W}}(\mathbf{X}_i^\pm)$ given $\mathbf{Z}_{[2n]}$. It then follows directly that the class-$f$-CMI bound is always tighter than the class-CMI bound. $\qquad\square$

### A.6 Extra Bound of Class-generalization Error using the Loss Pair $\mathbf{L}_i^\pm$:

**Theorem 7.** *(class-e-CMI) Assume that the loss $\ell(\hat{y}, y) \in [0, 1]$, then the class-generalization error of class $y$ in Definition 3 can be bounded as*

$$|\overline{\text{gen}_y}(P_{\mathbf{X}, \mathbf{Y}}, P_{\mathbf{W}|\mathbf{S}})| \leq \mathbb{E}_{\mathbf{Z}_{[2n]}}\Big[\frac{1}{n_{\mathbf{Z}_{[2n]}}^y} \sum_{i=1}^n \sqrt{2\max(\mathbb{1}_{\{\mathbf{Y}_i^- = y\}}, \mathbb{1}_{\{\mathbf{Y}_i^+ = y\}}) I_{\mathbf{Z}_{[2n]}}(\mathbf{L}_i^\pm; \mathbf{U}_i)}\Big]. \tag{40}$$

*Proof.* Similar to the proof of Theorems 2 and 3. Using Lemma 2 with $\mathbf{V} = \mathbf{L}_i^\pm$ and $h(\mathbf{V}, z_i) = \mathbf{L}_i$ in, we have

$$\mathbb{E}_{\mathbf{L}_i^\pm; \mathbf{U}_i|\mathbf{Z}_{[2n]} = z_{[2n]}}[g(f_{\mathbf{W}}(\mathbf{X}_i), \mathbf{U}_i, z_{[2n]})] \leq \sqrt{2\max(\mathbb{1}_{\{y_i^- = y\}}, \mathbb{1}_{\{y_i^+ = y\}}) I_{z_{[2n]}}(\mathbf{L}_i^\pm; \mathbf{U}_i)}. \tag{41}$$

Thus, taking expectation with respect to $\mathbf{Z}_{[2n]}$ yields the desired result

$$|\overline{\text{gen}_y}(P_{\mathbf{X}, \mathbf{Y}}, P_{\mathbf{W}|\mathbf{S}})| \leq \mathbb{E}_{\mathbf{Z}_{[2n]}}\Big[\frac{1}{n_{\mathbf{Z}_{[2n]}}^y} \sum_{i=1}^n \sqrt{2\max(\mathbb{1}_{\{\mathbf{Y}_i^- = y\}}, \mathbb{1}_{\{\mathbf{Y}_i^+ = y\}}) I_{\mathbf{Z}_{[2n]}}(\mathbf{L}_i^\pm; \mathbf{U}_i)}\Big]. \tag{42}$$

$\qquad\square$

### A.7 Proof of Theorem 4

**Theorem 4** (restated) Define $\Delta_y \mathbf{L}_i \triangleq \mathbb{1}_{\{y_i^- = y\}} \ell(f_{\mathbf{W}}(\mathbf{X}_i)^-, y_i^-) - \mathbb{1}_{\{y_i^+ = y\}} \ell(f_{\mathbf{W}}(\mathbf{X}_i)^+, y_i^+)$. Assume that the loss $\ell(\hat{y}, y) \in [0, 1]$ is bounded, then the class-generalization error of class $y$ in Definition 3 can be bounded as

$$|\overline{\text{gen}_y}(P_{\mathbf{X}, \mathbf{Y}}, P_{\mathbf{W}|\mathbf{S}})| \leq \mathbb{E}_{\mathbf{Z}_{[2n]}}\Big[\frac{1}{n_{\mathbf{Z}_{[2n]}}^y} \sum_{i=1}^n \sqrt{2 I_{\mathbf{Z}_{[2n]}}(\Delta_y \mathbf{L}_i; \mathbf{U}_i)}\Big]. \tag{43}$$

*Proof.* First, we notice that for a fixed realization $z_{[2n]}$, $\mathbb{1}_{\{y^{\mathbf{U}_i}=y\}}\ell(\mathbf{W}, z_i^{\mathbf{U}_i}) - \mathbb{1}_{\{y^{-\mathbf{U}_i}=y\}}\ell(\mathbf{W}, z_i^{-\mathbf{U}_i}) = \mathbf{U}_i(\mathbb{1}_{\{y_i^-=y\}}\ell(\mathbf{W}, z_i^-) - \mathbb{1}_{\{y_i^+=y\}}\ell(\mathbf{W}, z_i^+)) = \mathbf{U}_i \Delta_y \mathbf{L}_i$.

Next, let $(\overline{\Delta_y \mathbf{L}}_i, \overline{\mathbf{U}}_i)$ be an independent copy of $(\Delta_y \mathbf{L}_i; \mathbf{U}_i)$. Using the Donsker–Varadhan variational representation of KL divergence, we have $\forall \lambda \in \mathbb{R}$ and for every function $g$

$$I_{z_{[2n]}}(\Delta_y \mathbf{L}_i; \mathbf{U}_i) \geq \lambda \mathbb{E}_{\Delta_y \mathbf{L}_i, \mathbf{U}_i | \mathbf{Z}_{[2n]} = z_{[2n]}}[g(\Delta_y \mathbf{L}_i, \mathbf{U}_i, z_{[2n]})] \tag{44}$$
$$- \log \mathbb{E}_{\overline{\Delta_y \mathbf{L}}_i, \overline{\mathbf{U}}_i | \mathbf{Z}_{[2n]} = z_{[2n]}}[e^{\lambda g(\overline{\Delta_y \mathbf{L}}_i, \overline{\mathbf{U}}_i, z_{[2n]})}].$$

Next, let $g(\Delta_y \mathbf{L}_i, \mathbf{U}_i, z_{[2n]}) = \mathbf{U}_i \Delta_y \mathbf{L}_i$, and we have

$$\log \mathbb{E}_{\overline{\Delta_y \mathbf{L}}_i, \overline{\mathbf{U}}_i | \mathbf{Z}_{[2n]} = z_{[2n]}}[e^{\lambda g(\overline{\Delta_y \mathbf{L}}_i, \overline{\mathbf{U}}_i, z_{[2n]})}] = \log \mathbb{E}_{\overline{\Delta_y \mathbf{L}}_i, \overline{\mathbf{U}}_i | \mathbf{Z}_{[2n]} = z_{[2n]}}[e^{\lambda \overline{\mathbf{U}}_i \overline{\Delta_y \mathbf{L}}_i}]. \tag{45}$$

Note that $\mathbb{E}_{\overline{\mathbf{U}}_i}[\overline{\mathbf{U}}_i \overline{\Delta_y \mathbf{L}}_i] = 0$ and $\overline{\mathbf{U}}_i \in \{-1, +1\}$. Thus, by Hoeffding's Lemma, we have

$$\log \mathbb{E}_{\overline{\mathbf{L}}_i^\pm, \overline{\mathbf{U}}_i | \mathbf{Z}_{[2n]} = z_{[2n]}}[e^{\lambda g(\overline{\Delta_y \mathbf{L}}_i, \overline{\mathbf{U}}_i, z_{[2n]})}] \leq \log \mathbb{E}_{\overline{\Delta_y \mathbf{L}}_i | \mathbf{Z}_{[2n]} = z_{[2n]}}[e^{\frac{\lambda^2}{2} \overline{\Delta_y \mathbf{L}}_i^2}]. \tag{46}$$

As $\ell \in [0, 1]$, it follows that $\Delta_y \mathbf{L}_i \in [-1, 1]$, and $|\overline{\Delta_y \mathbf{L}}_i| \leq 1$. Thus,

$$\log \mathbb{E}_{\overline{\Delta_y \mathbf{L}}_i, \overline{\mathbf{U}}_i | \mathbf{Z}_{[2n]} = z_{[2n]}}[e^{\lambda g(\overline{\Delta_y \mathbf{L}}_i, \overline{\mathbf{U}}_i, z_{[2n]})}] \leq \frac{\lambda^2}{2}. \tag{47}$$

Replacing in equation 44, we have

$$I_{z_{[2n]}}(\Delta_y \mathbf{L}_i; \mathbf{U}_i) \geq \lambda \mathbb{E}_{\Delta_y \mathbf{L}_i, \mathbf{U}_i | \mathbf{Z}_{[2n]} = z_{[2n]}}[\mathbf{U}_i \Delta_y \mathbf{L}_i] - \frac{\lambda^2}{2}. \tag{48}$$

Therefore, $\forall \lambda \in \mathbb{R}$

$$\frac{\lambda^2}{2} - \lambda \mathbb{E}_{\Delta_y \mathbf{L}_i; \mathbf{U}_i | \mathbf{Z}_{[2n]} = z_{[2n]}}[g(\Delta_y \mathbf{L}_i, \mathbf{U}_i, z_{[2n]})] + I_{z_{[2n]}}(\Delta_y \mathbf{L}_i; \mathbf{U}_i) \geq 0. \tag{49}$$

The equation 49 is a non-negative parabola with respect to $\lambda$. Thus, its discriminant must be non-positive, which implies

$$\mathbb{E}_{\Delta_y \mathbf{L}_i; \mathbf{U}_i | \mathbf{Z}_{[2n]} = z_{[2n]}}[g(f_{\mathbf{W}}(\mathbf{X}_i), \mathbf{U}_i, z_{[2n]})] \leq \sqrt{2 I_{z_{[2n]}}(\Delta_y \mathbf{L}_i; \mathbf{U}_i)}. \tag{50}$$

Taking expectation with respect to $\mathbf{Z}_{[2n]}$ yields the desired result

$$|\overline{\text{gen}}_y(P_{\mathbf{X}, \mathbf{Y}}, P_{\mathbf{W}|\mathbf{S}})| \leq \mathbb{E}_{\mathbf{Z}_{[2n]}}\left[\frac{1}{n_{\mathbf{Z}_{[2n]}}^y} \sum_{i=1}^n \sqrt{2 I_{\mathbf{Z}_{[2n]}}(\Delta_y \mathbf{L}_i; \mathbf{U}_i)}\right]. \tag{51}$$

In the following, we will show that the $\Delta_y L$-CMI is always tighter than the class-$f$-CMI bound in Theorem 3. Due to the data processing inequality, we have $\mathbf{U} \to \mathbf{W} \to f_{\mathbf{W}}(\mathbf{X}_i^\pm) \to \Delta_y \mathbf{L}_i$ given $\mathbf{Z}_{[2n]}$. For a fixed $\mathbf{Z}_{[2n]}$, we have four different possible cases for each term in the sum:

1. If $y_i^- \neq y$ and $y_i^+ \neq y$: In this case, $\max(\mathbb{1}_{\{\mathbf{Y}_i^-=y\}}, \mathbb{1}_{\{\mathbf{Y}_i^+=y\}}) I_{\mathbf{Z}_{[2n]}}(f_{\mathbf{W}}(\mathbf{X}_i^\pm); \mathbf{U}_i) = 0$. On the other hand, we have $\Delta_y \mathbf{L}_i = 0$. Therefore, $I_{\mathbf{Z}_{[2n]}}(\Delta_y \mathbf{L}_i; \mathbf{U}_i) = 0 \leq \max(\mathbb{1}_{\{\mathbf{Y}_i^-=y\}}, \mathbb{1}_{\{\mathbf{Y}_i^+=y\}}) I_{\mathbf{Z}_{[2n]}}(f_{\mathbf{W}}(\mathbf{X}_i^\pm); \mathbf{U}_i)$.

2. If $y_i^- = y$ and $y_i^+ = y$: In this case, $\max(\mathbb{1}_{\{\mathbf{Y}_i^-=y\}}, \mathbb{1}_{\{\mathbf{Y}_i^+=y\}}) I_{\mathbf{Z}_{[2n]}}(f_{\mathbf{W}}(\mathbf{X}_i^\pm); \mathbf{U}_i) = I_{\mathbf{Z}_{[2n]}}(f_{\mathbf{W}}(\mathbf{X}_i^\pm); \mathbf{U}_i)$. Due to the data processing inequality, $I_{\mathbf{Z}_{[2n]}}(\Delta_y \mathbf{L}_i; \mathbf{U}_i) \leq I_{\mathbf{Z}_{[2n]}}(f_{\mathbf{W}}(\mathbf{X}_i^\pm); \mathbf{U}_i) = \max(\mathbb{1}_{\{\mathbf{Y}_i^-=y\}}, \mathbb{1}_{\{\mathbf{Y}_i^+=y\}}) I_{\mathbf{Z}_{[2n]}}(f_{\mathbf{W}}(\mathbf{X}_i^\pm); \mathbf{U}_i)$

3. If $y_i^+ \neq y$ and $y_i^- = y$: In this case, $\max(\mathbb{1}_{\{\mathbf{Y}_i^- = y\}}, \mathbb{1}_{\{\mathbf{Y}_i^+ = y\}}) I_{\mathbf{Z}_{[2n]}}(f_\mathbf{W}(\mathbf{X}_i^\pm); \mathbf{U}_i) = I_{\mathbf{Z}_{[2n]}}(f_\mathbf{W}(\mathbf{X}_i^\pm); \mathbf{U}_i)$ and $\Delta_y \mathbf{L}_i = \mathbf{L}_i^+$. As $\mathbf{W} \to f_\mathbf{W}(\mathbf{X}_i^\pm) \to \mathbf{L}_i^+$ is also a Markov chain, using the data processing inequality, we have $I_{\mathbf{Z}_{[2n]}}(\mathbf{L}_i^+; \mathbf{U}_i) \leq I_{\mathbf{Z}_{[2n]}}(f_\mathbf{W}(\mathbf{X}_i^\pm); \mathbf{U}_i)$ and thus $I_{\mathbf{Z}_{[2n]}}(\Delta_y \mathbf{L}_i; \mathbf{U}_i) \leq I_{\mathbf{Z}_{[2n]}}(f_\mathbf{W}(\mathbf{X}_i^\pm); \mathbf{U}_i)$.

4. If $y_i^+ = y$ and $y_i^- \neq y$: This case will be the same as the previous situation by swapping the $+$ and $-$.

Based on this discussion, we can conclude that

$$I_{\mathbf{Z}_{[2n]}}(\Delta_y \mathbf{L}_i; \mathbf{U}_i) \leq \max(\mathbb{1}_{\{\mathbf{Y}_i^- = y\}}, \mathbb{1}_{\{\mathbf{Y}_i^+ = y\}}) I_{\mathbf{Z}_{[2n]}}(f_\mathbf{W}(\mathbf{X}_i^\pm); \mathbf{U}_i) \tag{52}$$

$\Delta_y L$-CMI is always tighter than the class-$f$-CMI bound. $\qquad \square$

# B    Discussions on Definition 3

## B.1    Equivalence between Definition 1 and Definition 2

Here, we show the exact equivalence between Definition 1 and Definition 2.

$$\begin{aligned}
\widetilde{\mathrm{gen}}_y &\triangleq \frac{1}{n^y} \mathbb{E}_{\mathbf{Z}_{[2n]}} \Big[ \sum_{i=1}^n \mathbb{E}_{\mathbf{U}_i, \mathbf{W} | \mathbf{Z}_{[2n]}} \big[ \mathbb{1}_{\{Y_i^{-U_i} = y\}} \ell(\mathbf{W}, \mathbf{Z}_i^{-\mathbf{U}_i}) - \mathbb{1}_{\{Y_i^{U_i} = y\}} \ell(\mathbf{W}, \mathbf{Z}_i^{\mathbf{U}_i}) \big] \Big] \\
&= \frac{1}{nP(y)} \mathbb{E}_{\mathbf{Z}_{[2n]}} \mathbb{E}_{\mathbf{U}, \mathbf{W} | \mathbf{Z}_{[2n]}} \Big[ \sum_{i=1}^n \big[ \mathbb{1}_{\{Y_i^{-U_i} = y\}} \ell(\mathbf{W}, \mathbf{Z}_i^{-\mathbf{U}_i}) - \mathbb{1}_{\{Y_i^{U_i} = y\}} \ell(\mathbf{W}, \mathbf{Z}_i^{\mathbf{U}_i}) \big] \Big] \\
&= \frac{1}{n} \mathbb{E}_{\mathbf{U}, \mathbf{W}, \mathbf{Z}_{[2n]}} \Big[ \frac{1}{P(y)} \sum_{i=1}^n \big[ \mathbb{1}_{\{Y_i^{-U_i} = y\}} \ell(\mathbf{W}, \mathbf{Z}_i^{-\mathbf{U}_i}) - \mathbb{1}_{\{Y_i^{U_i} = y\}} \ell(\mathbf{W}, \mathbf{Z}_i^{\mathbf{U}_i}) \big] \Big] \\
&= \mathbb{E}_\mathbf{W} \mathbb{E}_{\mathbf{Z} \sim P_{\mathbf{Z}|y}} \big[ \ell(\mathbf{W}, \mathbf{Z}) \big] - \mathbb{E}_{\mathbf{Z}, \mathbf{W} | y} \big[ \ell(\mathbf{W}, \mathbf{Z}) \big] = \overline{\mathrm{gen}}_y(P_{\mathbf{X}, \mathbf{Y}}, P_{\mathbf{W} | \mathbf{S}}). 
\end{aligned} \tag{53}$$

Hence, class-generalization bounds based on this variants can be converted directly to standard generalization bounds by taking expectation over $y$, i.e., $\overline{\mathrm{gen}} = \mathbb{E}_\mathbf{Y}[\widetilde{\mathrm{gen}}_y]$.

## B.2    Other possible Definition by changing the super-samples setting

Another possible approach to study class-generalization error requires tweaking the super-sample setting as follows:

Let $\mathbf{Y}_{[n]} = \{\mathbf{Y}_1, \cdots, \mathbf{Y}_n\} \in \mathcal{Y}^n$ be a collection be a collection of $n$ i.i.d samples from $P_\mathbf{Y}$. Let $\mathbf{X}_{[2n]} = \{\mathbf{X}_1^\pm, \cdots, \mathbf{X}_n^\pm\} \in \mathcal{X}^{2n}$, such each pair $\mathbf{X}_i^\pm$ are drawn independently from the distribution $P_{\mathbf{X}|Y_i}$. Then the supersamples $\hat{\mathbf{Z}}_{[2n]} = (\mathbf{Z}_1^\pm, \cdots, \mathbf{Z}_n^\pm) \in \mathcal{Z}^{2n}$ is obtained such that $\mathbf{Z}_i^+ = (\mathbf{X}_i^+, \mathbf{Y}_i)$ and $\mathbf{Z}_i^- = (\mathbf{X}_i^-, \mathbf{Y}_i)$. The training data $\hat{\mathbf{Z}}_{[n]}^\mathbf{R} = (\mathbf{Z}_1^{\mathbf{R}_1}, \mathbf{Z}_2^{\mathbf{R}_2}, \cdots, \mathbf{Z}_n^{\mathbf{R}_n})$ is selected from the data $\hat{\mathbf{Z}}_{[2n]}$ where $\mathbf{R}_{[n]} = (\mathbf{R}_1, \cdots, \mathbf{R}_n) \in \{-1, 1\}^n$ is the vector composed of $n$ independent Rademacher random variables. Basically, $\mathbf{R}_i$ selects which sample from $\mathbf{Z}_i^\pm$ to be included in the training data and the other one for the test. The main difference compared to the CMI formulation in the main paper is that we construct our data such that for each $i \in 1, \cdots, n$, $\mathbf{Z}_i^+$ and $\mathbf{Z}_i^-$ are guaranteed to share the same label, while this is not necessarily the case for the prior formulations (Steinke & Zakynthinou, 2020; Zhou et al., 2022). Hence, in this setting, the two indicator functions in Definition 3 are equal and can be replaced with only one indicator function to define the class-generalization error.

The key issue with such a formulation lies in creating some dependency between the training set $\hat{\mathbf{Z}}_{[n]}^\mathbf{R}$ and the test set $\hat{\mathbf{Z}}_{[n]}^{-\mathbf{R}}$. Therefore, the difference between the loss evaluated using $\hat{\mathbf{Z}}_{[n]}^{-\mathbf{R}}$ and $\hat{\mathbf{Z}}_{[n]}^\mathbf{R}$ can no longer be interpreted as the true generalization error. Formally, in the main paper setup, by taking the expectation

over $\mathbf{Y}$ over Definition 2, we find

$$
\begin{aligned}
\mathbb{E}_{\mathbf{Z}_{[2n]},\mathbf{U},\mathbf{W}}\Big[\frac{1}{n}\sum_i\big(\ell(\mathbf{W},\mathbf{Z}_i^{-\mathbf{U}_i})-\ell(\mathbf{W},\mathbf{Z}_i^{\mathbf{U}_i})\big)\Big] &= \frac{1}{n}\sum_i\mathbb{E}_{\mathbf{Z}_i,\mathbf{U},\mathbf{W}}\Big[\big(\ell(\mathbf{W},\mathbf{Z}_i^{-\mathbf{U}_i})-\ell(\mathbf{W},\mathbf{Z}_i^{\mathbf{U}_i})\big)\Big] \\
&= \mathbb{E}_{\mathbf{W}}\mathbb{E}_{z\sim P_{\mathbf{Z}}}\big[\ell(\mathbf{W},\mathbf{Z})\big]-\mathbb{E}_{\mathbf{W},\mathbf{S}}\Big[\ell(\mathbf{W},\boldsymbol{S})\Big] \qquad (54)\\
&= \overline{\mathrm{gen}}(P_{\mathbf{Z}},P_{\mathbf{W}|\mathbf{S}}).
\end{aligned}
$$

This shows that the standard expected generalization error can be obtained by taking the expectation over $\mathbf{Y}$ for the class-generalization error. This would no longer be the case in the alternative formulation presented above. The equality in equation 54 can no longer be achieved due to the dependency between $\hat{\mathbf{Z}}_{[n]}^{-\mathbf{R}}$ and $\hat{\mathbf{Z}}_{[n]}^{\mathbf{R}}$. Hence, the bounds for the standard generalization error via the class-wise analysis (Section 5.1) can no longer be obtained directly. To see this, consider the following partition of the sample space $\mathcal{Z}^2 = \Omega \cup \bar{\Omega}$, such that $\Omega = \{z^{\pm}|z^{\pm}\in\mathcal{Z}^2, y^+ = y^-\}$ only contains sample pairs with the same label,

$$
\begin{aligned}
\overline{\mathrm{gen}}(P_{\mathbf{Z}},P_{\mathbf{W}|\mathbf{S}}) =& \mathbb{E}_{\hat{\mathbf{Z}}_{[2n]},\mathbf{U},\mathbf{W}}\Big[\frac{1}{n}\sum_i\big(\ell(\mathbf{W},\hat{\mathbf{Z}}_i^{-\mathbf{R}_i})-\ell(\mathbf{W},\hat{\mathbf{Z}}_i^{\mathbf{R}_i})\big)\Big] \\
=& P(\hat{\mathbf{Z}}_i^{\pm}\in\Omega)\mathbb{E}_{\hat{\mathbf{Z}}_{[2n]},\mathbf{U},\mathbf{W}}\Big[\frac{1}{n}\sum_i\big(\ell(\mathbf{W},\hat{\mathbf{Z}}_i^{-\mathbf{R}_i})-\ell(\mathbf{W},\hat{\mathbf{Z}}_i^{\mathbf{R}_i})\big)|\hat{\mathbf{Z}}_i^{\pm}\in\Omega\Big] \qquad (55)\\
&+ (1-P(\hat{\mathbf{Z}}_i^{\pm}\in\Omega))\mathbb{E}_{\hat{\mathbf{Z}}_{[2n]},\mathbf{U},\mathbf{W}}\Big[\frac{1}{n}\sum_i\big(\ell(\mathbf{W},\hat{\mathbf{Z}}_i^{-\mathbf{R}_i})-\ell(\mathbf{W},\hat{\mathbf{Z}}_i^{\mathbf{R}_i})\big)|\hat{\mathbf{Z}}_i^{\pm}\in\bar{\Omega}\Big].
\end{aligned}
$$

The above decomposition of the generalization error contains two terms: The first term measures the generalization gap between samples with the same label. Thus, it can be interpreted as a 'within-class generalization error.' The second term measures the generalization error between the samples with different labels. It can be interpreted as a 'between-class generalization error.' Taking the expectation over $\mathbf{Y}$ of the class-generalization error, in the setting considered with $\hat{\mathbf{Z}}_{[2n]}$, will only include the first term in equation 55. Thus, we consider the class-generalization error given by Definitions 2-3 instead of the setting discussed here.

## C  Additional Empirical Results

### C.1  Experiment Setup

Here, we fully describe the experimental setup used in the main body of the paper. In our study, we focused on balanced datasets such as CIFAR-10/CIFAR100, where each class is represented equally, to ensure that the observed class-generalization disparities are not confounded by class imbalance. This balance allows us to attribute disparities in class-generalization error to intrinsic properties of the model and data distribution rather than to skewed class proportions.

We use the same setup as in Harutyunyan et al. (2021), where the code is publicly available[3]. For every number of training data $n$, we run $m_1$ number of Monte-Carlo trials, i.e., we select $m_1$ different $2n$ samples from the original dataset.Then, for each $z_{[2n]}$, we draw $m_2$ different train/test splits, i.e., $m_2$ random realizations of $\mathbf{U}$. In total, we have $m_1m_2$ experiments. We report the mean and standard deviation on the $m_1$ results. For the CIFER10 experiments, we select $m_1 = 2$ and $m_2 = 20$. For its noisy variant, we select $m_1 = 5$ and $m_2 = 15$. For both datasets, we use ResNet50 pre-trained on ImageNet. The training is conducted for 40 epochs using SGD with a learning rate of 0.01 and a batch size of 256.

### C.2  Full Class-generalization Error vs. Standard Generalization Results on CIFAR10

As a supplement to Figure 1 (b), we plot the standard generalization error along with the class-generalization error of all the classes of CIFAR10 in Figure 6. Consistent with Section 1, we observe significant variability in generalization performance across different classes.

---

[3]https://github.com/hrayrhar/f-CMI/tree/master

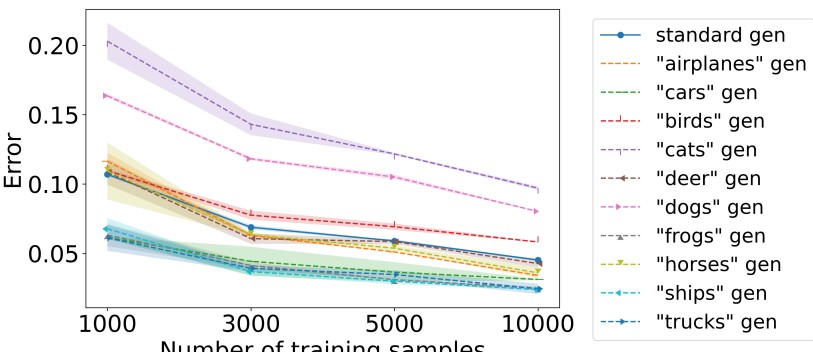

Figure 6: The standard generalization error and the generalization error relative for all classes on CIFAR10 as a function of the number of training data.

## C.3  Numerical Results for All Classes of CIFAR10 and Its Noisy Variants

In Figure 7, we present the empirical evaluation of our bounds on all the classes of CIFAR10. Moreover, we generate the scatter plot between the class-generalization error and the class-$f$-CMI bound. We note that similar to the class-$\Delta L_y$ results in Figure 3, our bound scales linearly with the true class-generalization error. The results of noisy CIFAR10 with clean validation, presented in Figure 8, are also consistent with these findings. We also experimented with noisy CIFAR10 with noise added to both the train and validations. The results are presented in 9. Our bounds, in this case, are able to capture the behavior of the class-generalization error for the different classes.

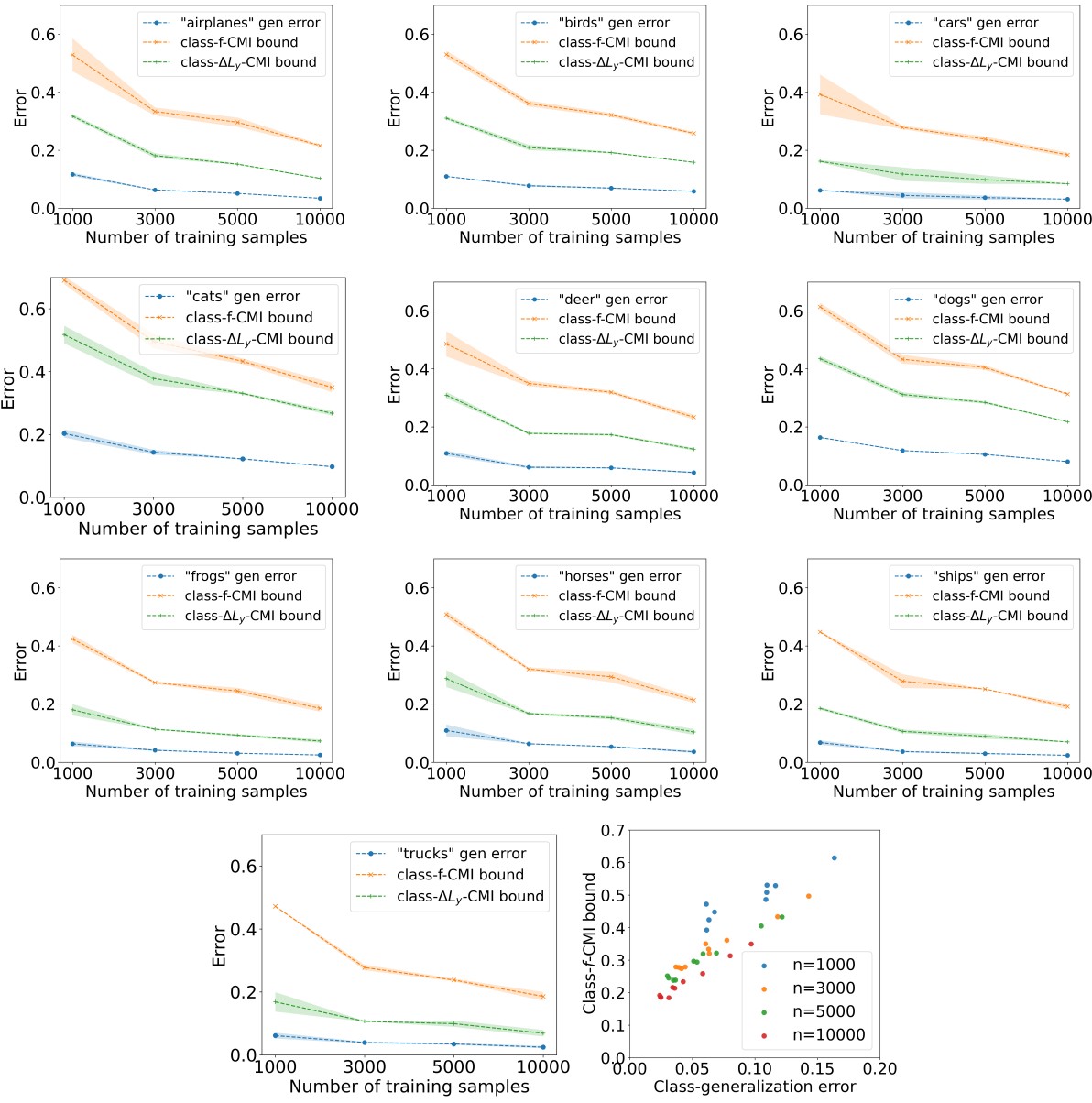

Figure 7: Class-wise generalization on the ten classes of CIFAR10 and the scatter plot between class-generalization error and the class-$f$-CMI bound in Theorem 3.

## C.4 Numerical results for CIFAR100

Here, we present the empirical evaluation of our bounds on a more complex dateset. namely CIFAR100. We use the same experimental setup as for CIFAR10 (in Section C.3). In Figure 10, we generate the scatter-plots between the class-generalization error and both the class-$f$-CMI and the class-$\Delta L_y$ bounds under different number of samples. As can be see, our bounds, especially the class-$\Delta L_y$ bound, are indeed linearly correlated with the class-generalization error and can efficiently predict its behavior, even for a dataset with high number of classes.

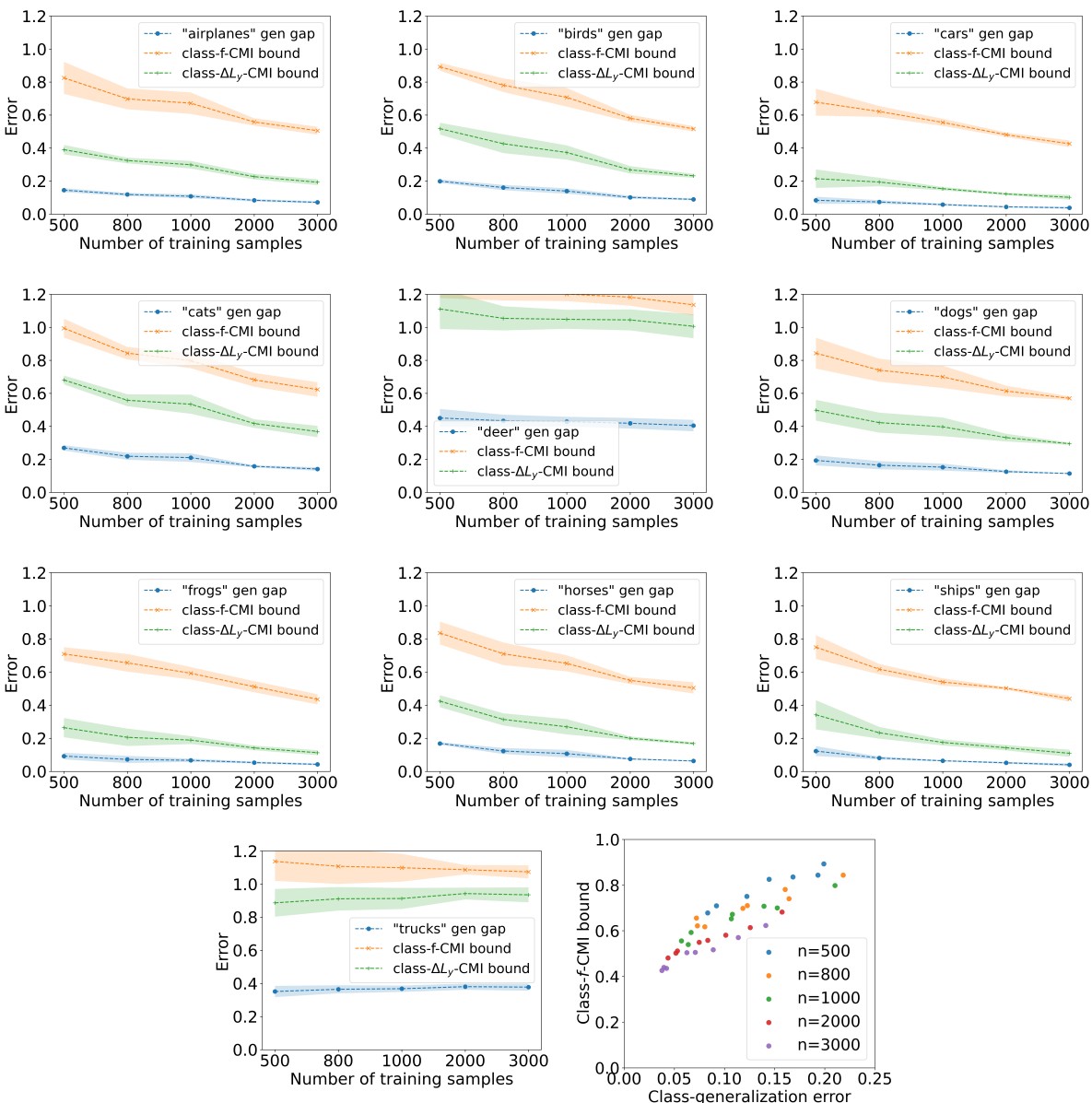

Figure 8: Class-wise generalization on the ten classes of **noisy** CIFAR10 (clean validation) and the scatter plot between class-generalization error and the class-$f$-CMI bound in Theorem 3.

## C.5 Numerical results for SVM & Decision Trees

Here, we present the empirical evaluation of our bounds for two classic ML approaches, namely SVM and Random Forest Classifier on MNIST dataset. The main results for both approaches are presented in Figures 11 and 12, respectively. As can be seen, the results for both approaches are consistent with the neural networks' experiments further confirming the ability of our bounds to capture the complex behavior of class-generalization.

## C.6 Extra Results: Expected Recall & Specificity Generalization

In the special case of binary classification with the 0-1 loss, the class-generalization errors studied within this paper correspond to generalization in terms of recall and specificity:

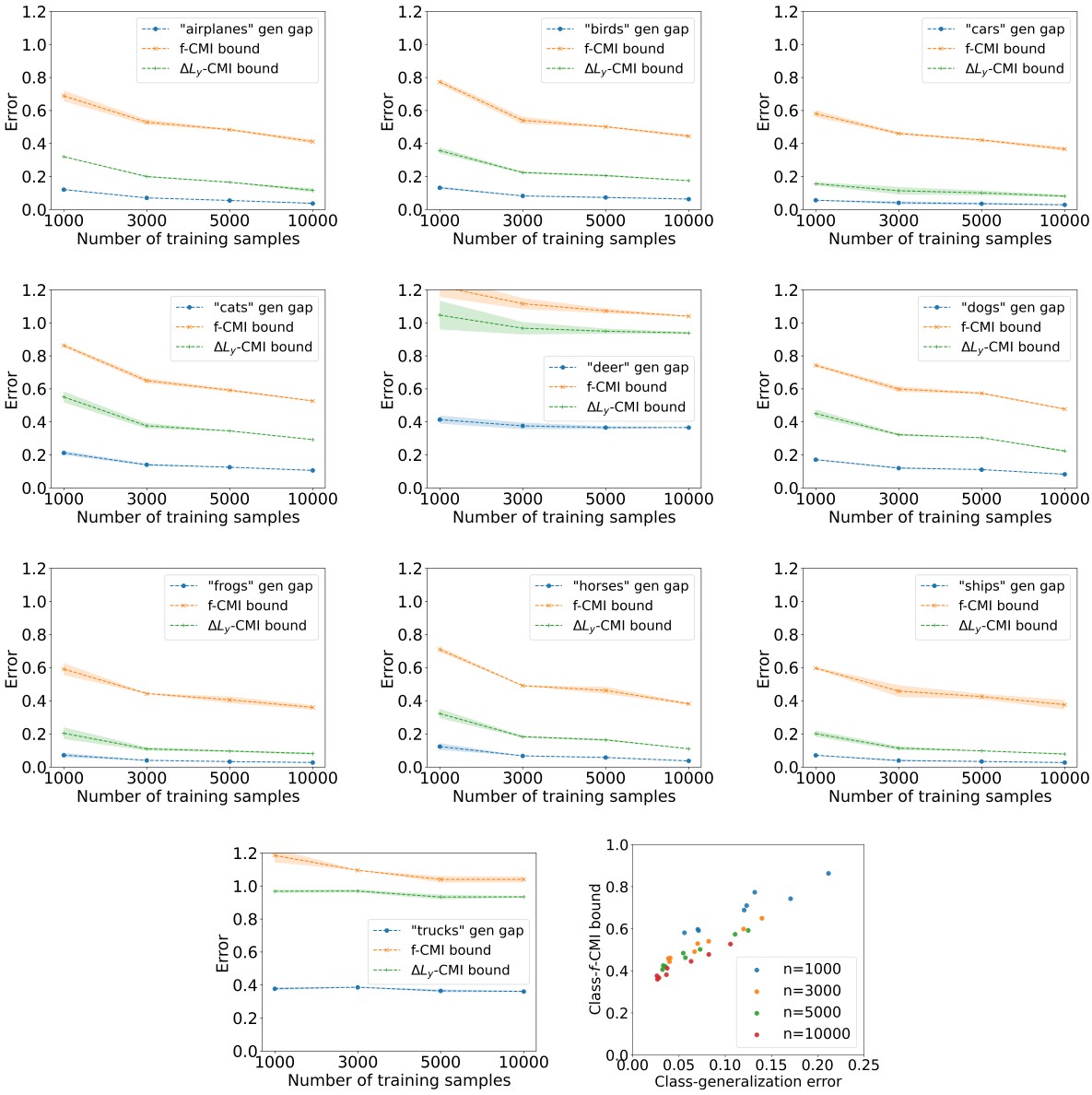

Figure 9: Class-wise generalization on the ten classes of **noisy** CIFAR10 (noise added to both train and validation) and the scatter plot between class-generalization error and the class-$f$-CMI bound in Theorem 3.

$expected\ recall - empirical\ recall = gen_p$, where $p$ is the positive class.
$expected\ specificity - empirical\ specificity = gen_n$, where $n$ is the negative class.

Standard Generalization bounds (Xu & Raginsky, 2017; Harutyunyan et al., 2021) provide theoretical certificates for learning algorithms regarding classification error/accuracy. However, in several ML applications, e.g., an imbalanced binary dataset, accuracy/error are not considered good performance metrics. For example, consider the binary classification problem of detecting a rare cancer type. While analyzing the model's generalization error is important, we might be more interested in understanding generalization in terms of recall, as that is more critical in this case. Standard theoretical bounds (Neyshabur et al., 2017; Wu et al., 2020; Wang & Mao, 2023) do not provide any insights for such metrics.

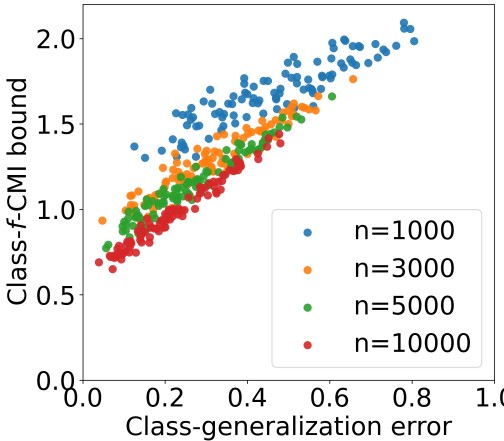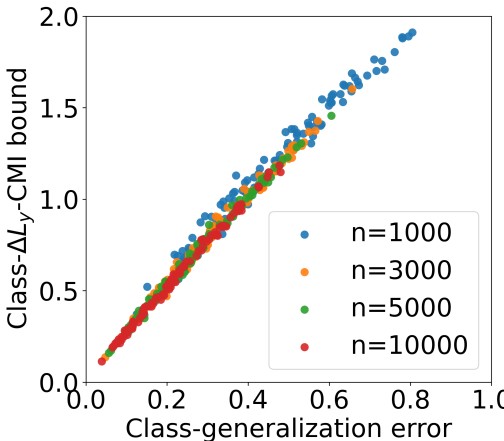

Figure 10: Experimental results of class-generalization error and our bounds in Theorems 3 and 4 for CIFAR100. We provide the scatter plots between the bound in the true class-generalization error and both: i) our bound in Theorem 3 (left); ii) our bound in Theorem 4 (right) for the different classes for CIFAR100.

The tools developed in this paper can be used to close this gap and allow us to understand generalization for recall and specificity theoretically.

We conduct an experiment of binary MNIST (digit 4 vs. digit 9), similar to Harutyunyan et al. (2021). $m_1$ and $m_2$ discussed in Section C.1 are selected to be $m_1 = 5$ and $m_2 = 30$. Empirical results for this case are presented in Figure 4. As can be seen in the Figure, our bounds efficiently estimate the expected recall and specificity errors.

### C.7 Extra Results: Generalization Certificates with Sensitive Attributes

Here, we present the empirical evaluation of our bounds for generalization with sensitive attributes using two datasets, namely using the **COMPAS** (Larson et al., 2016) and **Adult** (Kohavi & Becker, 1996) dataset. The main results for the different sub-population are presented in Figures 5 and 13, respectively. As can be seen, the results are consistent with the previous experiments further confirming the ability of our bounds to capture the complex behavior of sub-population based generalization

### C.8 Class-specific Gradient Noise Improves Class-generalization

In this paper, we studied the phenomenon that the generalization errors for the same model can differ significantly among different classes by introducing and exploring the concept of "class-generalization error." This provides a first theoretical step toward understanding this puzzling phenomenon. Our results show that the mutual information between the model and the class data can be used as a proxy for this class-generalization error. In other words, when the mutual information between the class samples and the model's parameters is high (high memorization), the model overfits this class (poor generalization) and vice versa.

Therefore, to improve the model's generalization performance for a specific important class $y$, our results suggest reducing the MI/CMI between the training samples with class $y$ and the model weights/output. A straightforward approach to achieve this is by adding noise to the gradient updates during training when a sample in the batch has the label $y$.

We validate and confirm the effectiveness of this idea with two learning scenarios with target classes: (i) "cars" and (ii) "cats" from the CIFAR10 dataset. We use random Gaussian noise with zero mean and variance of 0.005. The results are reported in Table 1. As can be seen, using this simple regularization consistently reduces the generalization error and the test error of these classes in both scenarios. This further confirms

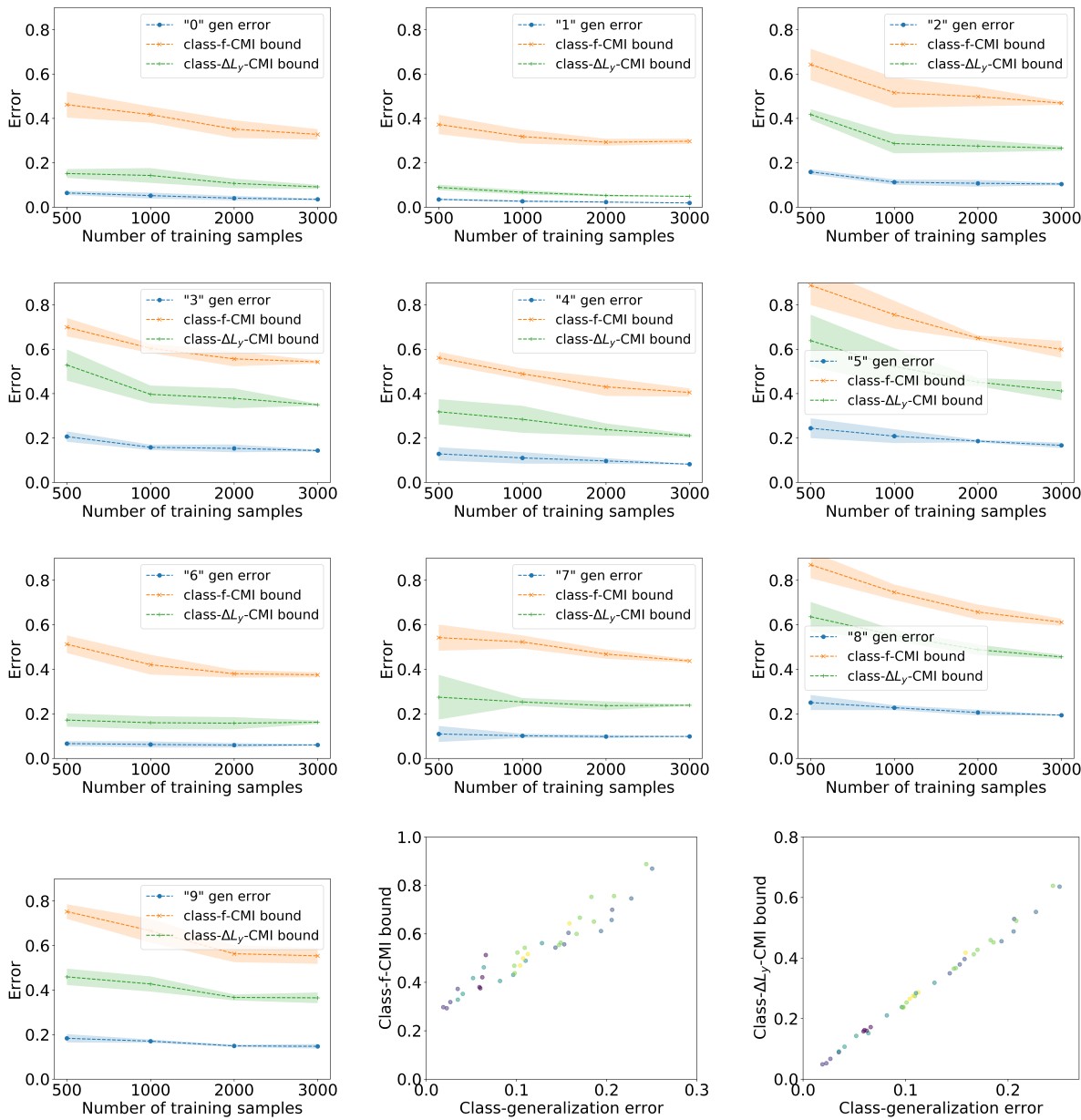

Figure 11: SVM: Class-wise generalization on the ten classes of MNIST and the scatter plot between class-generalization error and the class-$f$-CMI bound in Theorem 3.

the theoretical findings of our paper and provides some insights into potential approaches to improve class generalization in the desired applications.

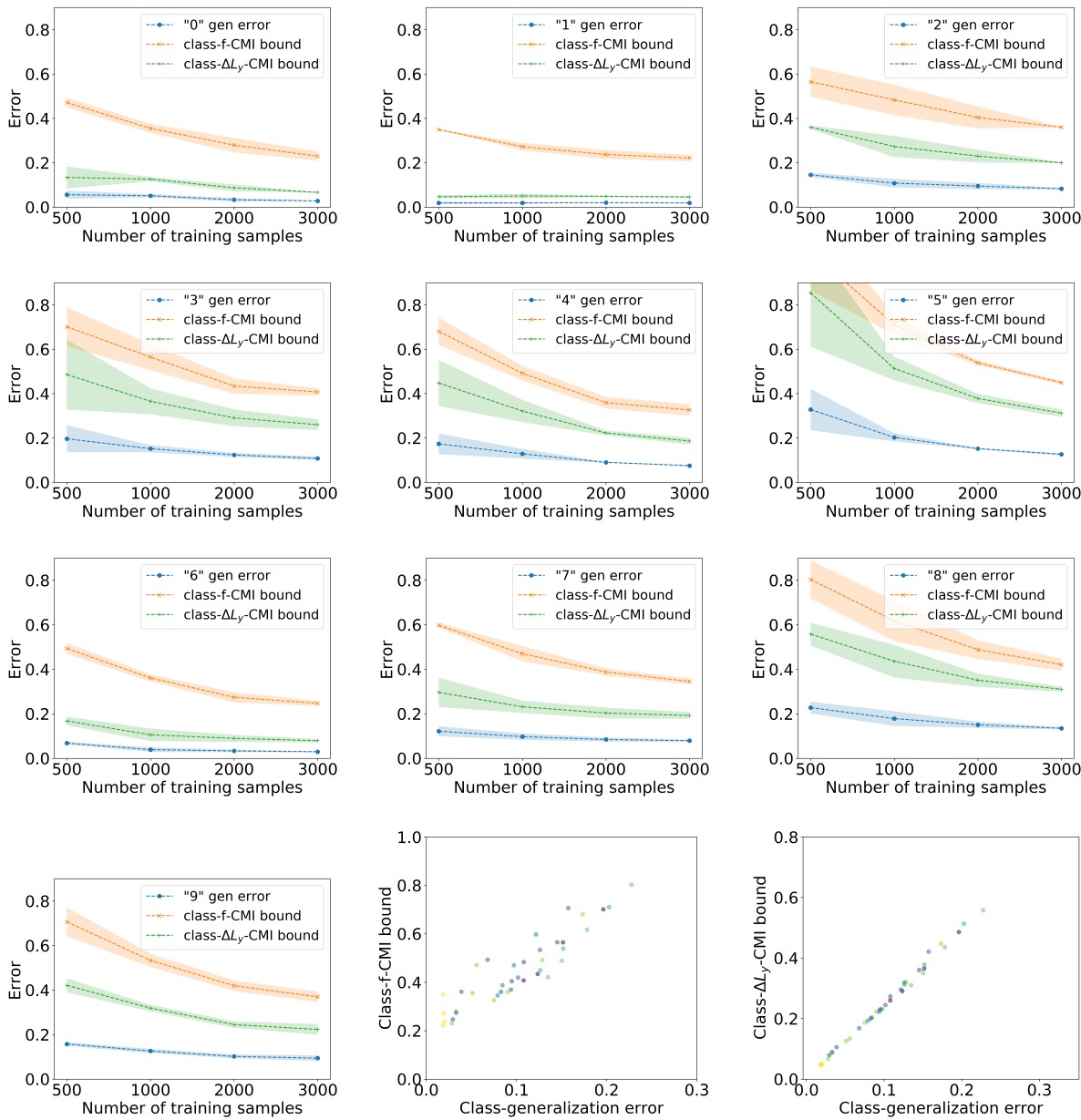

Figure 12: Random Forest: Class-wise generalization on the ten classes of MNIST and the scatter plot between class-generalization error and the class-$f$-CMI bound in Theorem 3.

# D  Details for Results in Section 5

## D.1  Full Details of Section 5.1: Standard Generalization Error

**Corollary 1** (restated) Assume that for every $y \in \mathcal{Y}$, the loss $\ell(\overline{\mathbf{W}}, \overline{\mathbf{X}}, y)$ is $\sigma_y$ sub-Gaussian under $P_{\overline{\mathbf{W}}} \otimes P_{\overline{\mathbf{X}}|\overline{\mathbf{Y}}=y}$, then

$$|\overline{\mathrm{gen}}(P_{\mathbf{X},\mathbf{Y}}, P_{\mathbf{W}|\mathbf{S}})| \leq \frac{1}{n} \sum_{i=1}^{n} \mathbb{E}_Y \sqrt{2\sigma_{\mathbf{Y}}^2 D(P_{\mathbf{W},\mathbf{X}_i|\mathbf{Y}_i=y} || P_{\mathbf{W}} \otimes P_{\mathbf{X}_i|\mathbf{Y}_i=y})}. \tag{56}$$

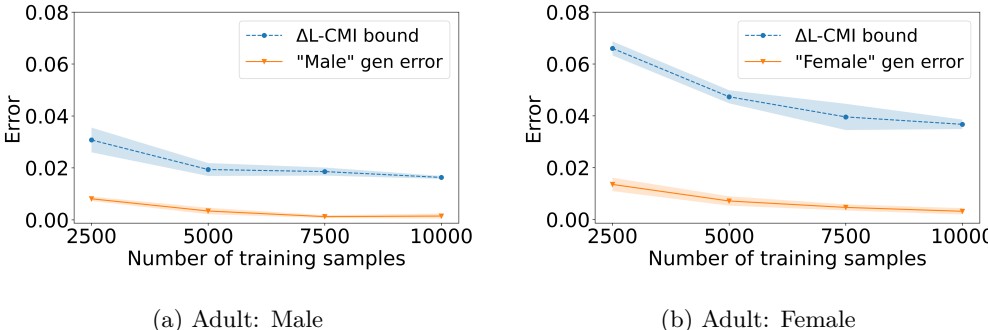

(a) Adult: Male

(b) Adult: Female

Figure 13: Empirical evaluation of our bounds for generalization with sensitive attributes using using the **Adult** dataset.

Table 1: average class-classification errors and generalization errors of the target class in the different scenarios.

| method | "cars" | | "cats" | |
|---|---|---|---|---|
| | $\text{test}_y$ | $\text{gen}_y$ | $\text{test}_y$ | $\text{gen}_y$ |
| ERM | 3.91% | 3.91% | 12.27% | 12.24% |
| Ours | 3.55% | 3.54% | 12.11% | 12.09% |

*Proof.* The generalization error can be written as

$$\overline{\text{gen}}(P_{\mathbf{X},\mathbf{Y}}, P_{\mathbf{W}|\mathbf{S}}) = \frac{1}{n} \sum_{i=1}^{n} \left( \mathbb{E}_{\mathbf{W},\overline{\mathbf{Z}}}[\ell(\mathbf{W}, \overline{\mathbf{Z}})] - \mathbb{E}_{\mathbf{W},\mathbf{Z}_i}[\ell(\mathbf{W}, \mathbf{Z}_i)] \right). \tag{57}$$

As the loss $\ell$ is $\sigma_{\mathbf{Y}}$ sub-Gaussian, using Theorem 1, we have

$$\mathbb{E}_{P_{\mathbf{W},\mathbf{X}|\mathbf{Y}=y}}[\ell(\mathbf{W}, \mathbf{X}, \mathbf{Y})] - \mathbb{E}_{P_{\mathbf{W}} \otimes P_{\mathbf{X}|\mathbf{Y}=y}}[\ell(\overline{\mathbf{W}}, \overline{\mathbf{X}}, \overline{\mathbf{Y}})] \le \sqrt{2\sigma_y^2 D(P_{\mathbf{W},\mathbf{X}|\mathbf{Y}=y} || P_{\mathbf{W}} \otimes P_{\mathbf{X}|\mathbf{Y}=y})}. \tag{58}$$

Taking the expectation over $\mathbf{Y}$ in both sides, we have

$$\mathbb{E}_{P_{\mathbf{W},\mathbf{X},\mathbf{Y}}}[\ell(\mathbf{W}, \mathbf{X}, \mathbf{Y})] - \mathbb{E}_{P_{\mathbf{W}} \otimes P_{\mathbf{X},\mathbf{Y}}}[\ell(\overline{\mathbf{W}}, \overline{\mathbf{X}}, \overline{\mathbf{Y}})] \le \mathbb{E}_Y \sqrt{2\sigma_{\mathbf{Y}}^2 D(P_{\mathbf{W},\mathbf{X}|\mathbf{Y}=y} || P_{\mathbf{W}} \otimes P_{\mathbf{X}|\mathbf{Y}=y})}. \tag{59}$$

Applying equation 59 on each term of equation 57 for each $\mathbf{Z}_i$ completes the proof.

$\square$

**Comparison between Corollary 1 and bounds in Bu et al. (2020)** In the case of standard loss sub-Gaussianity assumption, i.e., $\sigma_y = \sigma$ is independent of $y$, it is possible to show that the bound in 1 is tighter than the bound in Bu et al. (2020). This is because

$$\mathbb{E}_{P_Y} \sqrt{2\sigma^2 D(P_{\mathbf{W},\mathbf{X}|\mathbf{Y}} || P_{\mathbf{W}} \otimes P_{\mathbf{X}|\mathbf{Y}})} = \mathbb{E}_{P_Y} \sqrt{2\sigma^2 \mathbb{E}_{P_{\mathbf{W},\mathbf{X}|\mathbf{Y}}} \log \frac{P_{\mathbf{W},\mathbf{X}|\mathbf{Y}} P_Y}{P_{\mathbf{W}} \otimes P_{\mathbf{X}|\mathbf{Y}} P_Y}}$$

$$= \mathbb{E}_{P_Y} \sqrt{2\sigma^2 \mathbb{E}_{P_{\mathbf{W},\mathbf{X}|\mathbf{Y}}} \log \frac{P_{\mathbf{W},\mathbf{X},\mathbf{Y}}}{P_{\mathbf{W}} \otimes P_{\mathbf{X},\mathbf{Y}}}} \le \sqrt{2\sigma^2 I(\mathbf{W}; \mathbf{Z})} \tag{60}$$

where the last inequality comes from Jensen's inequality. This shows that class-wise analysis can be used to derive tighter generalization bounds.

In the supersamples setting, extending the results in Theorems 2, 3, and 4 into standard generalization bounds is not strightforward, as we do not have $\overline{\text{gen}} = \mathbb{E}_{\mathbf{Y}}[\overline{\text{gen}}_y]$. However, it is still possible to show that $\overline{\text{gen}} \leq \mathbb{E}_{\mathbf{Y}}[\overline{\text{gen}}_y]$ and hence the bounds in Theorems 2, 3, and 4 can be used to derive class-dependent bounds for the standard generalization error in the supersample setting. For example, in Corollary 2, we provide such an extension of Theorem 4.

by taking the expectation over $y \sim P_{\mathbf{Y}}$. In Corollaries 4, 5, 6, and 2, we provide such an extension of Theorems 2, 3, 7, and 4, respectively.

**Corollary 2**(restated) Assume that the loss $\ell(\hat{y}, y) \in [0, 1]$, then

$$|\overline{\text{gen}}(P_{\mathbf{X},\mathbf{Y}}, P_{\mathbf{W}|\mathbf{S}})| \leq \mathbb{E}_{\mathbf{Y}}\left[\mathbb{E}_{\mathbf{Z}_{[2n]}}\left[\frac{1}{n^{\mathbf{Y}}}\sum_{i=1}^{n}\sqrt{2I_{\mathbf{Z}_{[2n]}}(\Delta_{\mathbf{Y}}\mathbf{L}_i; \mathbf{U}_i)}\right]\right]. \tag{61}$$

**Corollary 4.** *(extra result) Assume that the loss $\ell(\hat{y}, y) \in [0, 1]$, then*

$$|\overline{\text{gen}}(P_{\mathbf{X},\mathbf{Y}}, P_{\mathbf{W}|\mathbf{S}})| \leq \mathbb{E}_{\mathbf{Y}}\left[\mathbb{E}_{\mathbf{Z}_{[2n]}}\left[\frac{1}{n^{\mathbf{Y}}}\sum_{i=1}^{n}\sqrt{2\max(\mathbb{1}_{\mathbf{Y}_i^- = \mathbf{Y}}, \mathbb{1}_{\mathbf{Y}_i^+ = \mathbf{Y}})I_{\mathbf{Z}_{[2n]}}(\mathbf{W}; \mathbf{U}_i)}\right]\right]. \tag{62}$$

**Corollary 5.** *(extra result) Assume that the loss $\ell(\hat{y}, y) \in [0, 1]$, then*

$$|\overline{\text{gen}}(P_{\mathbf{X},\mathbf{Y}}, P_{\mathbf{W}|\mathbf{S}})| \leq \mathbb{E}_{\mathbf{Y}}\left[\mathbb{E}_{\mathbf{Z}_{[2n]}}\left[\frac{1}{n^{\mathbf{Y}}}\sum_{i=1}^{n}\sqrt{2\max(\mathbb{1}_{\mathbf{Y}_i^- = \mathbf{Y}}, \mathbb{1}_{\mathbf{Y}_i^+ = \mathbf{Y}})I_{\mathbf{Z}_{[2n]}}(f_{\mathbf{W}}(\mathbf{X}_i^{\pm}); \mathbf{U}_i)}\right]\right]. \tag{63}$$

**Corollary 6.** *(extra result) Assume that the loss $\ell(\hat{y}, y) \in [0, 1]$, then*

$$|\overline{\text{gen}}(P_{\mathbf{X},\mathbf{Y}}, P_{\mathbf{W}|\mathbf{S}})| \leq \mathbb{E}_{\mathbf{Y}}\left[\mathbb{E}_{\mathbf{Z}_{[2n]}}\left[\frac{1}{n^{\mathbf{Y}}}\sum_{i=1}^{n}\sqrt{2\max(\mathbb{1}_{\mathbf{Y}_i^- = \mathbf{Y}}, \mathbb{1}_{\mathbf{Y}_i^+ = \mathbf{Y}})I_{\mathbf{Z}_{[2n]}}(\mathbf{L}_i^{\pm}; \mathbf{U}_i)}\right]\right]. \tag{64}$$

### D.2 Full Details of Section 5.2: Sub-task Problem

Consider a supervised learning problem where the machine learning model $f_{\mathbf{W}}(\cdot)$, parameterized with $w \in \mathcal{W}$, is obtained with a training dataset $S$ consisting of $n$ i.i.d samples $z_i = (x_i, y_i) \in \mathcal{X} \times \mathcal{Y} \triangleq \mathcal{Z}$ generated from distribution $P_{\mathbf{XY}}$. The quality of the model with parameter $w$ is evaluated with a loss function $\ell: \mathcal{W} \times \mathcal{Z} \to \mathbb{R}^+$.

For any $w \in \mathcal{W}$, the population risk is defined as follows

$$L_P(w) = \mathbb{E}_{P_{\mathbf{X},\mathbf{Y}}}[\ell(w, \mathbf{X}, \mathbf{Y})]. \tag{65}$$

and the empirical risk is:

$$L_{E_P}(w, S) = \frac{1}{n}\sum_{i=1}^{n}\ell(w, x_i, y_i). \tag{66}$$

Here, we are interested in the subtask problem, which is a special case of distribution shift, i.e., the test performance of the model $w$ is evaluated using a specific subset of classes $\mathcal{A} \subset \mathcal{Y}$ of the source distribution $P_{\mathbf{XY}}$. Thus, the target distribution $Q_{\mathbf{XY}}$ is defined as $Q_{\mathbf{XY}}(x, y) = \frac{P_{\mathbf{XY}}(x,y)\mathbb{1}_{\{y \in \mathcal{A}\}}}{P_{\mathbf{Y}}(y \in \mathcal{A})}$. The population risk on the target domain $Q$ of the subtask problem is

$$L_Q(w) = \mathbb{E}_{Q_{\mathbf{X},\mathbf{Y}}}[\ell(w, \mathbf{X}, \mathbf{Y})]. \tag{67}$$

A learning algorithm can be modeled as a randomized mapping from the training set $S$ onto a model parameter $w \in \mathcal{W}$ according to the conditional distribution $P_{\mathbf{W}|S}$. The expected generalization error on the subtask problem is the difference between the population risk of $Q$ and the empirical risk evaluated using all samples from $S$:

$$\overline{\text{gen}}_{Q, E_P} = \mathbb{E}_{P_{\mathbf{W},\mathbf{s}}}[L_Q(\mathbf{W}) - L_{E_P}(\mathbf{W}, \mathbf{S})], \tag{68}$$

where the expectation is taken over the joint distribution $P_{\mathbf{W},\mathbf{S}} = P_{\mathbf{W}|S} \otimes P_{\mathbf{Z}}^n$.

The generalization error defined above can be decomposed as follows:

$$\overline{\text{gen}}_{Q,E_P} = \mathbb{E}_{P_{\mathbf{W}}}[L_Q(\mathbf{W}) - L_P(\mathbf{W})] + \mathbb{E}_{P_{\mathbf{W},\mathbf{s}}}[L_P(\mathbf{W}) - L_{E_P}(\mathbf{W},\mathbf{S})]. \tag{69}$$

The first term quantifies the gap of the population risks in two different domains, and the second term is the source domain generalization error. Assuming that loss is $\sigma$-sub-Gaussian under $P_{\mathbf{Z}}$, it is shown in Wang & Mao (2022) that the first term can be bounded using the KL divergence between $P$ and $Q$:

$$\mathbb{E}_{P_{\mathbf{W}}}[L_Q(w) - L_P(w)] \leq \sqrt{2\sigma^2 D(Q\|P)}. \tag{70}$$

The second term can be bounded using the standard mutual information approach in Xu & Raginsky (2017) as

$$\mathbb{E}_{P_{\mathbf{W},\mathbf{s}}}[L_P(\mathbf{W}) - L_{E_P}(\mathbf{W},\mathbf{S})] \leq \sqrt{2\sigma^2 I(\mathbf{W};\mathbf{S})}. \tag{71}$$

Thus, the generalization error of the subtask problem can be bounded as follows:

$$\overline{\text{gen}}_{Q,E_P} \leq \sqrt{2\sigma^2 D(Q\|P)} + \sqrt{2\sigma^2 I(\mathbf{W};\mathbf{S})}. \tag{72}$$

Obtaining tighter generalization error bounds for the subtask problem is straightforward using our class-wise generalization bounds. In fact, the generalization error bound of the subtask can be obtained by taking the expectation of $\mathbf{Y} \sim Q_{\mathbf{Y}}$.

Using Jensen's inequality, we have $|\overline{\text{gen}}_{Q,E_Q}| = |\mathbb{E}_{\mathbf{Y} \sim Q_{\mathbf{Y}}}[\overline{\text{gen}}_{\mathbf{Y}}]| \leq \mathbb{E}_{\mathbf{Y} \sim Q_{\mathbf{Y}}}[|\overline{\text{gen}}_{\mathbf{Y}}|]$. Thus, we can use the results from Section 3 to obtain tighter bounds.

**Theorem 5** (subtask-$\Delta L_y$-CMI) (restated) Assume that the loss $\ell(w,x,y) \in [0,1]$ is bounded, Then the subtask generalization error defined in 14 can be bounded as

$$|\overline{\text{gen}}_{Q,E_Q}| \leq \mathbb{E}_{\mathbf{Y} \sim Q_{\mathbf{Y}}}\left[\mathbb{E}_{\mathbf{Z}_{[2n]}}\left[\frac{1}{n^{\mathbf{Y}}}\sum_{i=1}^{n}\sqrt{2I_{\mathbf{Z}_{[2n]}}(\Delta_{\mathbf{Y}}\mathbf{L}_i;\mathbf{U}_i)}\right]\right].$$

Similarly, we can also extend the result of Theorem 2 and 3 to the subtask as follows:

**Theorem 8.** *(subtask-CMI) (extra result) Assume that the loss $\ell(w,x,y) \in [0,1]$ is bounded, then the subtask generalization error defined in 14 can be bounded as*

$$|\overline{\text{gen}}_{Q,E_Q}| \leq \mathbb{E}_{\mathbf{Y} \sim Q_{\mathbf{Y}}}\left[\mathbb{E}_{\mathbf{Z}_{[2n]}}\left[\frac{1}{n^{\mathbf{Y}}}\sum_{i=1}^{n}\sqrt{2\max(\mathbb{1}_{\mathbf{Y}_i^-=\mathbf{Y}},\mathbb{1}_{\mathbf{Y}_i^+=\mathbf{Y}})I_{\mathbf{Z}_{[2n]}}(\mathbf{W};\mathbf{U}_i)}\right]\right].$$

**Theorem 9.** *(subtask-f-CMI) (extra result) Assume that the loss $\ell(w,x,y) \in [0,1]$ is bounded, then the subtask generalization error defined in 14 can be bounded as*

$$|\overline{\text{gen}}_{Q,E_Q}| \leq \mathbb{E}_{\mathbf{Y} \sim Q_{\mathbf{Y}}}\left[\mathbb{E}_{\mathbf{Z}_{[2n]}}\left[\frac{1}{n^{\mathbf{Y}}}\sum_{i=1}^{n}\sqrt{2\max(\mathbb{1}_{\mathbf{Y}_i^-=\mathbf{Y}},\mathbb{1}_{\mathbf{Y}_i^+=\mathbf{Y}})I_{\mathbf{Z}_{[2n]}}(f_{\mathbf{W}}(\mathbf{X}_i^{\pm});\mathbf{U}_i)}\right]\right].$$

### D.2.1 Empirical Validation of Bounds for Sub-task problem

We conduct an experiment of the CIFAR10 dataset, similar to Section 4. We design two subtask problems with this dataset. In the first scenario, referred to here by subtask1, we consider the target distribution to be composed of the two classes "airplanes" and "cars". Whereas in the second scenario, referred to here by subtask2, we construct the target distribution using three classes, namely "airplanes", "cars" and "birds".

$m_1$ and $m_2$ discussed in Section C.1 are selected to be $m_1 = 2$ and $m_2 = 15$. Empirical results of the bounds in Theorems 9 and 5 are presented in Figure 14. As can be seen in the Figure, our bounds efficiently estimate the generalization errors for the subtask problem.

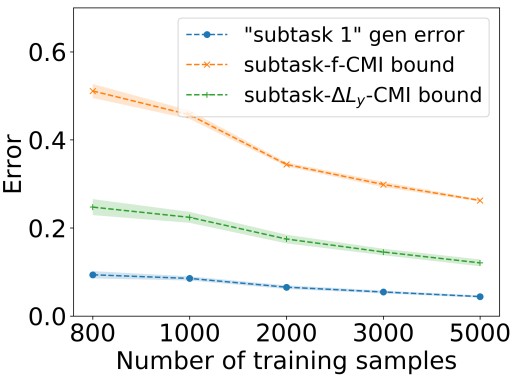 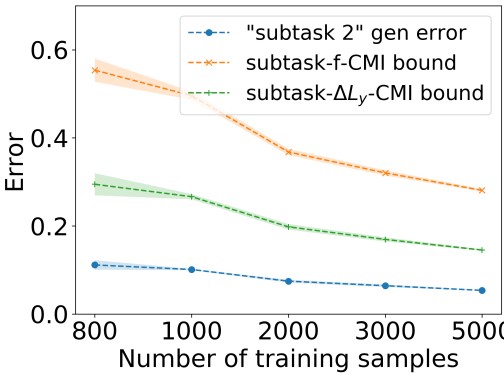

Figure 14: Generalization performance on two subtask1 ("airplanes" and "cars") and subtask2 "airplanes", "cars" and "birds") problems constructed using CIFAR10 dataset for different training sample sizes.

### D.3 Full Details of Section 5.3: Generalization Certificates with Sensitive Attributes

**Theorem 6** (restated) Given $t \in \mathcal{T}$, assume that the loss $\ell(\mathbf{W}, \mathbf{Z})$ is $\sigma$ sub-Gaussian under $P_{\overline{\mathbf{W}}} \otimes P_{\overline{\mathbf{Z}}}$, then the attribute-generalization error of the sub-population $\mathbf{T} = t$, as defined in 4, can be bounded as follows:

$$|\overline{\text{gen}_t}(P_{\mathbf{X}, \mathbf{Y}}, P_{\mathbf{W}|\mathbf{S}})| \leq \sqrt{2\sigma^2 D(P_{\mathbf{W}|\mathbf{Z}} \otimes P_{\mathbf{Z}|\mathbf{T}=t} || P_{\mathbf{W}} \otimes P_{\mathbf{Z}|\overline{\mathbf{T}}=t})}. \tag{73}$$

*Proof.* We have

$$\overline{\text{gen}_t}(P_{\mathbf{X}, \mathbf{Y}}, P_{\mathbf{W}|\mathbf{S}}) = \mathbb{E}_{P_{\overline{\mathbf{W}}} \otimes P_{\overline{\mathbf{Z}}|\mathbf{T}=t}}[\ell(\overline{\mathbf{W}}, \overline{\mathbf{Z}})] - \mathbb{E}_{P_{\mathbf{W}|\mathbf{Z}} \otimes P_{\mathbf{Z}|\mathbf{T}=t}}[\ell(\mathbf{W}, \mathbf{Z})]. \tag{74}$$

Using the Donsker–Varadhan variational representation of the relative entropy, we have

$$D(P_{\mathbf{W}|\mathbf{Z}} \otimes P_{\mathbf{Z}|\mathbf{T}=t} || P_{\mathbf{W}} \otimes P_{\mathbf{Z}|\mathbf{T}=t}) \geq \mathbb{E}_{P_{\mathbf{W}|\mathbf{Z}} \otimes P_{\mathbf{Z}|\mathbf{T}=t}}[\lambda \ell(\mathbf{W}, \mathbf{Z})] - \log \mathbb{E}_{P_{\overline{\mathbf{W}}} \otimes P_{\overline{\mathbf{Z}}|\overline{\mathbf{T}}=t}}[e^{\lambda \ell(\overline{\mathbf{W}}, \overline{\mathbf{Z}})}], \forall \lambda \in \mathbb{R}. \tag{75}$$

On the other hand, we have:

$$\log \mathbb{E}_{P_{\overline{\mathbf{W}}} \otimes P_{\overline{\mathbf{Z}}|\overline{\mathbf{T}}=t}} \left[ e^{\lambda \ell(\overline{\mathbf{W}}, \overline{\mathbf{Z}}) - \lambda \mathbb{E}[\ell(\overline{\mathbf{W}}, \overline{\mathbf{Z}})]} \right]$$
$$= \log \mathbb{E}_{P_{\overline{\mathbf{W}}} \otimes P_{\overline{\mathbf{Z}}|\overline{\mathbf{T}}=t}} \left[ e^{\lambda \ell(\overline{\mathbf{W}}, \overline{\mathbf{Z}})} e^{-\lambda \mathbb{E}[\ell(\overline{\mathbf{W}}, \overline{\mathbf{Z}})]} \right) \right]$$
$$= \log \mathbb{E}_{P_{\overline{\mathbf{W}}} \otimes P_{\overline{\mathbf{Z}}|\overline{\mathbf{T}}=t}} [e^{\lambda \ell(\overline{\mathbf{W}}, \overline{\mathbf{Z}})}] - \lambda \mathbb{E}_{P_{\overline{\mathbf{W}}} \otimes P_{\overline{\mathbf{Z}}|\overline{\mathbf{T}}=t}} [\ell(\overline{\mathbf{W}}, \overline{\mathbf{Z}})].$$

Using the sub-Gaussian assumption, we have

$$\log \mathbb{E}_{P_{\overline{\mathbf{W}}} \otimes P_{\overline{\mathbf{Z}}|\overline{\mathbf{T}}=t}} [e^{\lambda \ell(\overline{\mathbf{W}}, \overline{\mathbf{Z}})}] \leq \lambda \mathbb{E}_{P_{\overline{\mathbf{W}}} \otimes P_{\overline{\mathbf{Z}}|\overline{\mathbf{T}}=t}} (\ell(\overline{\mathbf{W}}, \overline{\mathbf{Z}})) + \frac{\lambda^2 \sigma^2}{2}. \tag{76}$$

By replacing in equation 75, we have

$$D(P_{\mathbf{W}|\mathbf{Z}} \otimes P_{\mathbf{Z}|\mathbf{T}=t} || P_{\mathbf{W}} \otimes P_{\mathbf{Z}|\mathbf{T}=t}) \geq \lambda \big( \mathbb{E}_{P_{\mathbf{W}|\mathbf{Z}} \otimes P_{\mathbf{Z}|\mathbf{T}=t}}[\ell(\mathbf{W}, \mathbf{Z})] - \mathbb{E}_{P_{\overline{\mathbf{W}}} \otimes P_{\overline{\mathbf{Z}}|\overline{\mathbf{T}}=t}}[\ell(\overline{\mathbf{W}}, \overline{\mathbf{Z}})] \big) - \frac{\lambda^2 \sigma}{2}. \tag{77}$$

Thus, we have:

$$D(P_{\mathbf{W}|\mathbf{Z}} \otimes P_{\mathbf{Z}|\mathbf{T}=t} || P_{\mathbf{W}} \otimes P_{\mathbf{Z}|\mathbf{T}=t}) - \lambda (\mathbb{E}_{P_{\mathbf{W}|\mathbf{Z}} \otimes P_{\mathbf{Z}|\mathbf{T}=t}}[\ell(\mathbf{W}, \mathbf{Z})] - \mathbb{E}_{P_{\overline{\mathbf{W}}} \otimes P_{\overline{\mathbf{Z}}|\overline{\mathbf{T}}=t}}[\ell(\overline{\mathbf{W}}, \overline{\mathbf{Z}})])$$
$$+ \lambda^2 \sigma^2 \geq 0, \forall \lambda \in \mathbb{R}. \tag{78}$$

equation 78 is a non-negative parabola with respect to $\lambda$. Thus, its discriminant must be non-positive. This implies

$$|\mathbb{E}_{P_{\mathbf{W}|\mathbf{Z}} \otimes P_{\mathbf{Z}|\mathbf{T}=t}}[\ell(\mathbf{W}, \mathbf{Z})] - \mathbb{E}_{P_{\overline{\mathbf{W}}} \otimes P_{\overline{\mathbf{Z}}|\overline{\mathbf{T}}=t}}[\ell(\overline{\mathbf{W}}, \overline{\mathbf{Z}})]| \leq \sqrt{2\sigma^2 D(P_{\mathbf{W}|\mathbf{Z}} \otimes P_{\mathbf{Z}|\mathbf{T}=t} || P_{\mathbf{W}} \otimes P_{\mathbf{Z}|\mathbf{T}=t})}, \qquad (79)$$

which completes the proof. $\qquad\square$

