# OpenReview forum: "Class-wise Generalization Error: an Information-Theoretic analysis"
_TMLR — Accepted by TMLR_

### Review · Reviewer_uLVn · 2025-03-26

**Summary Of Contributions:**

This work introduces the concept of class-generalization error, a measure designed to quantify how supervised learning algorithms generalize for each individual class within a data distribution. Using a class-dependent approach, the authors derive class-specific generalization bounds by adapting standard information-theoretic generalization analyses to this novel class-specific setting. The following results are important:
1. Bounds within the mutual information setting:
     1. The class generalization error of a given class can be upper bounded using the KL divergence between the joint distribution of weights and data, and their marginal distribution, both conditioned on the class occurence (Theorem 1). By taking the expectation over all classes, this yields a tighter global generalization bound than the traditional individual sample bound (Corollary 1).
2. Bounds within the conditional mutual information setting:
     1. Using the super-sample setting, the authors derive class-specific generalization upper bounds involving the conditional mutual information between the weights and the selection variables conditioned on a super-sample. (Theorem 2)
     2. The authors further derive tighter bounds that are easier to estimate and allow for black-box estimation. The bounds involve respectively the conditional mutual information between the model's output and the selection variables (Theorem 3) and between the difference of loss functions evaluated at the examples containing the class in question and selection variables, providing a tighter bound (Theorem 4).
    3. Using these bounds, the authors derive a generic generalization bound by taking the expectation over all classes (Corollary 2).
3. Extension to sub-task problems: The authors obtain tighter generalization bounds using the class-wise generalization results they have derived
4. Generalization error bounds with respect to sensitive attributes: A really interesting extension of the results developed by the authors is the derivation of attribute-wise generalization error, which provided a great potential for deriving bias-reducing methods in machine learning.

**Audience:**

Yes

**Claims And Evidence:**

Yes

**Requested Changes:**

The following changes would strengthen the work but are not critical to securing our recommendation for acceptance:
 1. The experiments were conducted on CIFAR10 and CIFAR100, which are both balanced datasets. It would be a good comparison to run the same experiments on a different dataset, in order to unveil the effect of class imbalance on the estimated bounds, especially to see if the bounds are still linearly correlated with the true error.
2. No insight is provided into why noisy CIFAR10 is used alongside CIFAR10.
3. The notation is sometimes heavy for the reader, and this is understandably due to the mathematical objects involved in the CMI setting. A more careful expression of which variables are conditioned on $Y=y$ is recommended.
4. In the paragraph following the statement of Theorem 2, it is stated that when the model memorizes training data, the class-generalization error will be large. However, theorem 2 only implies that the upper bound on the class-generalization error becomes large, and there is a number of works highlighting that memorization can sometimes improve generalization.
5. In the paragraph preceding section 5, point i) states (paraphrased): "one can use our bound to predict which classes will generalize better than others" but there is no argument in the paper justifying this claim.
6. The results in appendix C.7 show a slight improvement and are not sufficient to corroborate the claim "Such a result provides a new perspective on improving class-generalization by reducing MI/CMI. The initial empirical results in Appendix C.7 show that this is a promising research direction to mitigate the class generalization disparity."

**Strengths And Weaknesses:**

### Strenghts:
The problem under study is well motivated, and the theoretical tools derived provide insights into the varying performance of supervised learning algorithms on different classes, and can be further used in other machine learning problems as detailed in section 5.
Experiments further demonstrate that both the class-$f$-CMI and class-$\Delta$-CMI bounds capture well the behaviour of the class generalization error bounds.

---

> ### Author Response · Authors · 2025-05-07
>
> We thank the Reviewer for his positive comments.
>
>
> > The experiments were conducted on CIFAR10 and CIFAR100, which are both balanced datasets. It would be a good comparison to run the same experiments on a different dataset, in order to unveil the effect of class imbalance on the estimated bounds, especially to see if the bounds are still linearly correlated with the true error.
>
> We appreciate the reviewer’s suggestion to evaluate our approach on an imbalanced dataset. While class imbalance is indeed an important consideration, we have no reason to believe that our bounds would not apply in such settings, as the bounding technique holds for any learning algorithm and dataset. More importantly, our study focuses on class-wise generalization behavior, such as that illustrated in Figure 1, which is particularly puzzling because it appears in balanced datasets like CIFAR-10 and CIFAR-100.  For this reason, we selected these benchmarks for our experiments. We consider exploring the impact of class imbalance and how ERM-like learning algorithms affect class-wise generalization a valuable direction for future work.
>
> > No insight is provided into why noisy CIFAR10 is used alongside CIFAR10.
>
> The primary motivation behind introducing 5% label noise is to demonstrate that even minor corruptions in the data distribution can significantly impact class-wise generalization behavior. We intentionally selected a moderate noise level to avoid severely impairing overall learning, while still revealing how sensitive class-generalization can be. As the results show, even small perturbations lead to noticeable changes in class-level generalization, underscoring the need for more fine-grained generalization analysis. We have now clarified this motivation in the figure caption and the main text in the revision.
>
> > In the paragraph following the statement of Theorem 2, it is stated that when the model memorizes training data, the class-generalization error will be large. However, theorem 2 only implies that the upper bound on the class-generalization error becomes large, and there is a number of works highlighting that memorization can sometimes improve generalization.
>
> We thank the reviewer for pointing this out. We agree that Theorem 2 formally characterizes an upper bound on class-generalization error, not the error itself. However, in the information-theoretic learning literature, such upper bounds are widely used as proxies for generalization behavior. Therefore, analyzing what causes the bound to increase offers meaningful insights into what may lead to worse generalization in practice. We acknowledge that using the term memorization may have caused confusion. What we intended to highlight is that when the learned model becomes highly sensitive to specific training examples (as captured by increased mutual information between the training data and the model), the bound—and potentially the generalization error—tends to grow. We have revised the discussion following Theorem 2 to clarify this point and avoid conflating memorization with generalization degradation.
>
> > In the paragraph preceding section 5, point i) states (paraphrased): "one can use our bound to predict which classes will generalize better than others" but there is no argument in the paper justifying this claim.
>
> We appreciate the reviewer’s observation. As noted in our response to the previous comment, our claim is grounded in the standard practice of using upper bounds as proxies for true generalization performance, a well-established approach in learning theory. Furthermore, our claim is empirically supported by the strong linear correlation observed between the bound and the actual class-wise generalization error, as shown in Figure 3. This suggests that our bound can effectively serve as a practical indicator of which classes are more likely to generalize well.
>
> > The results in appendix C.7 show a slight improvement and are not sufficient to corroborate the claim "Such a result provides a new perspective on improving class-generalization by reducing MI/CMI. The initial empirical results in Appendix C.7 show that this is a promising research direction to mitigate the class generalization disparity."
>
> We thank the reviewer for this observation. The experiment in Appendix C.7 is intended only as a proof of concept, illustrating how our theoretical insights can potentially be translated into practice. While the observed improvements are modest, they demonstrate the potential of using our theory to inform algorithmic design. We believe that developing more effective regularization techniques that reduce the CMI terms could lead to more substantial performance gains.

---

### Review · Reviewer_Ejav · 2025-04-04

**Summary Of Contributions:**

This paper introduces the notion of class-wise generalization error—a measure of how well a learning algorithm generalizes for each class individually, as opposed to the standard average generalization error across all classes. The authors propose information-theoretic bounds on class-wise generalization using KL divergence and Conditional Mutual Information (CMI), including a tighter variant using the so-called $\Delta\_y L$-CMI.

Key contributions are as follows:

- The derivation of upper bounds for class-generalization error using KL divergence and multiple CMI-based quantities.
- They show the connection between the standard average generalization error and proposed bounds.
- Empirical validation on MNIST / CIFAR10 and a noisy variant, showing the tightness and class-dependence of the proposed bounds.
- Extension of the theoretical tools to standard generalization bounds, subtask problems, and fairness analysis (e.g., sensitive attributes).

**Audience:**

Yes

**Broader Impact Concerns:**

I believe this work does not raise any ethical concerns, as it is a theoretical study focused on the fundamental issue of generalization error in the context of machine learning.

**Claims And Evidence:**

Yes

**Requested Changes:**

To address the shortcomings identified earlier, I believe it is important to clarify and correct the points summarized below. If I have misunderstood anything, I would greatly appreciate your guidance in identifying the inaccuracies.

**Discuss in more depth the practical implications of class-generalization bounds:**
- When is it better to analyze class-wise error instead of standard generalization in practice?
- As a more speculative question: if one can estimate the class-wise generalization gap (even via an upper bound), could this information be used to adjust or correct the predicted probabilities for specific classes in terms of generalization? For instance, classes with large generalization gaps might systematically exhibit overconfidence, and class-wise bounds may provide a way to recalibrate such predictions.

**Strengthen the fairness-related discussion in Section 5.3:**
- Can the proposed bounds provide meaningful fairness guarantees in practice?
- How do they relate to existing fairness metrics or frameworks?

**Move or summarize key empirical results from the appendix into the main paper, especially:**
- The evaluation on classical models (Appendix C.5).
- The connection between class-generalization error and recall/specificity (Appendix C.6).

Optionally, include experiments on a more diverse benchmark (e.g., UCI benchmarks) to highlight the flexibility of the proposed framework.

**Strengths And Weaknesses:**

First and foremost, I would like to express my sincere appreciation for the authors’ efforts in developing this paper.
I would also like to note that the following review is written with the understanding that TMLR places a strong emphasis on accuracy, persuasiveness, and clear evidence, as well as capturing the interest of readers, rather than solely focusing on novelty or impact.


## Strengths

- The derivation of KL and CMI-based bounds for class-wise generalization error is rigorous and well-structured.
- The bounds are empirically validated across different scenarios, including noisy data and traditional ML models.
- The proposed framework is shown to extend naturally to other settings such as subtask generalization and fairness-related concerns.


## Weakness

- **Limited discussion of practical applicability**: The practical significance of estimating class-wise generalization error remains underexplored.
  - For instance, while Section 5.3 presents a theoretical extension to attribute-wise generalization for fairness evaluation, the paper does not investigate how meaningful or actionable the proposed bounds are in actual fairness-sensitive settings. Personally, I find the idea of evaluating generalization through the lens of fairness more compelling than simply observing that deep networks generalize unequally across labels---a phenomenon that might be anticipated to some extent, considering that neural networks are known to be poorly calibrated in classification (as suggested by prior calibration literature such as [Guo et al., 2017](https://arxiv.org/pdf/1706.04599)).
  A deeper discussion of how class-wise generalization could be used as a tool for fairness auditing would greatly strengthen the paper’s contribution.

- Underutilization of empirical insights in the appendix: Appendix C.5 (on traditional models) and C.6 (on recall/specificity connections) contain promising results that are not sufficiently discussed in the main text. Especially in the context of fairness, these experiments could have served as strong support for the practical value of the proposed approach.

- Limited diversity of experiments: While the theoretical focus of the paper is clear and appreciated, the empirical results are mostly limited to CIFAR10, MNIST, and their noisy variants. Including experiments on diverse benchmarks—such as fairness-oriented datasets, or even non-vision domains—would help showcase the generality and broader applicability of the proposed framework.


## Concerns regarding the interest of TMLR readers

This paper is likely to interest TMLR readers working on generalization theory, information-theoretic analysis, and fair ML.
However, to reach a broader audience, the paper would benefit from a more concrete discussion of the practical motivations and downstream benefits of class-wise generalization error analysis—particularly in real-world applications such as fairness.

---

> ### Author Response · Authors · 2025-05-07
>
> We thank the Reviewer for his positive comments.
>
> > When is it better to analyze class-wise error instead of standard generalization in practice?
>
> Thank you for this insightful question. From a practical perspective,  class-wise analysis becomes particularly relevant in scenarios involving high-stakes decisions tied to specific classes (e.g., medical diagnosis). In such cases, relying on averaged generalization errors can obscure critical performance gaps. As you can see from all the Figures in our paper, our bounds effectively track the generalization error of each case.  We have now added a discussion in the introduction to highlight this point.
>
> > As a more speculative question: if one can estimate the class-wise generalization gap (even via an upper bound), could this information be used to adjust or correct the predicted probabilities for specific classes in terms of generalization? ...
>
> Thank you for this interesting question. We don’t see how our bounds (or generalization bounds in general) can be used to calibrate models at the current stage, as generalization error and calibration are fundamentally distinct concepts. From our perspective, class-wise generalization bounds can potentially serve as indicators for identifying classes with high overfitting. However, we agree that studying the link between class-generalization error and class-calibration error (for specific loss functions, e.g., cross entropy) is an interesting future research direction.
>
> > Strengthen the fairness-related discussion in Section 5.3: \
> Can the proposed bounds provide meaningful fairness guarantees in practice? \
> How do they relate to existing fairness metrics or frameworks?
>
> Thank you for appreciating the extension of our results to the fairness setting. We have now expanded the discussion in Section 5.3 to better clarify the distinction between our approach and existing fairness literature. Most prior works on fairness generalization focus on population risks—that is, they study whether a model performs similarly across sensitive groups (e.g., male vs. female), and quantify population risk disparities  in terms of metrics such as Equalized Odds or Demographic Parity. In contrast, our work focuses on generalization error (vs population risk). Specifically, our bounds are conditioned on a fixed sensitive attribute (e.g., male) and quantify how well the model generalizes from training to testing for that group. Hence, our framework does not directly enforce fairness constraints, it provides a complementary analysis on models’ generalization for each subpopulation.
>
> We have now added a experiment that evaluates our proposed bounds in a fairness context using two datasets (COMPAS and Adult). The results are presented in Figure 5 and Figure 13.
>
> > Move or summarize key empirical results from the appendix into the main paper
>
> We appreciate this suggestion. Due to page limits, we are unable to include the extensive results and analysis in the main paper. However, we have now incorporated additional discussion of the results in Appendices C.5 and C.6 into the main paper, providing a more concise overview of these findings. Additionally, we have move empirical results for: i) sensitivity and recall and ii) fairness to the main body of the paper.

---

> ### Comment · Reviewer_Ejav · 2025-05-22
> **Acknowledgement**
>
> Dear authors,
>
> I apologize my late reply, and thank the authors for their detailed and thoughtful response. The revisions and additions—especially the clarified fairness discussion and the added experiments on the COMPAS and Adult datasets—substantially improve the practical relevance of the paper. While some limitations remain regarding empirical breadth and direct applicability, I believe the paper makes a solid theoretical contribution and addresses the raised concerns adequately.
>
> Sincerely,
> --Reviewer Ejav

---

### Review · Reviewer_UMUd · 2025-04-22

**Summary Of Contributions:**

The paper tackles addresses the fact that neural networks in classification often have different performances for different classification classes and that standard generalization bounds do not account for this. The paper therefore presents a class-generalization error and presents tractable versions in the super-sampling setting using the Conditional Mutual Information framework. These are evaluated using the CIFAR10 dataset; the Appendix provides further analysis with other classical ML methods (SVM and Decision trees).

**Audience:**

Yes

**Broader Impact Concerns:**

None.

**Claims And Evidence:**

Yes

**Requested Changes:**

Note that I have bundled minor and major changes together below. I should also note that I am not directly involved in this specific sub-field of work, so my review is based on how someone like that would read the paper; if there are any sub-field specific conventions or knowledge that I have missed, then please let me know in the rebuttal.

### Introduction

- 'Therefore, reasoning only concerning the standard generalization error (red curve) cannot capture this class-wise behavior.' -> 'Therefore, reasoning that only concerns the standard generalization error (red curve)**, which averages over the classes,** cannot capture this class-wise behavior.' : I think it would be good to reword as such to emphasize that standard generalization errors average the error over the different classes.

- Figure 1: Could you add figure titles to the left and right cases? Furthermore, why not use subfigure titles, such as Figure 1a and Figure 1b, rather than (left) and (right)?

- Figure 1: Could the caption say how many repetitions were done?

- Figure 1: Can the markers be made larger; they are quite difficult to see at the moment.

- 'this phenomenon using information-theoretic generalization bounds, as they are both data-dependent and algorithm-dependent' -> 'this phenomenon using information-theoretic generalization bounds, as **such bounds** are both data-dependent and algorithm-dependent': This makes it clear that 'they' in the original sentence refers to the generalization bounds.

- Figure 1: I think it would better to show the uncertainty as error bars, rather than interpolating.

- First paragraph page 2:

- 'When comparing the results on both datasets, it is worth noting that the generalization error of the same class “trucks” behaves significantly differently on the two datasets. This suggests that class-wise generalization highly depends on factors beyond the class itself, including the data distribution, the learning algorithm, and the number of training samples.' : I am not sure that I fully agree that the second sentence can be concluded from the first. My understanding is that the only difference between the two figures is that a small amount of noise was added; so how can this imply that _the learning algorithm_ affects the generalization? I am not saying that it doesn't, but that Figure 1 doesn't imply this.

- Notations: Why are random variables in bold?

- Notations: Can you write a sentence about that the intuition behind the _disintegrated conditional mutual information_ is? Since this is an important part of Theorem 2, I believe it would be useful for the reader to understand if they hadn't come across the term before.

### Related Work
- Information-theoretic generalization error bounds: Can a sentence or two be written that explains what _Information-theoretic generalization error bounds_ are? Specifically I am interested in why they were developed and what benefit they bring.

### Class-Generalization Error
- I found it slightly confusing that the notation $\mathbf{Z}$ exists. Why not just use $\mathbf{X}, \mathbf{Y}$? I think this is mainly because the text often uses some variation of $x, y$, $\mathbf{X} | y$, etc, and for just the specific case of $\mathbf{X}, \mathbf{Y}$, it changes to $\mathbf{Z}$. I am of the opinion that this isn't necessary. Furthermore, what is the difference between $\mathbf{S}$ and $\mathbf{Z}$?

### Empirical Evaluations
- Based on this section and the Introduction, it is not clear to me why the 5% label noise dataset is significant. As in, why was the case of added noise considered, and why this level of noise? Why not a large amount on noise?

- Figure 3: Same as Figure 1; please put titles to each sub-figure. Perhaps a grid-like notation can be used, where each column and row has a title on the top and left edges.

**Strengths And Weaknesses:**

### Strengths
- The idea and contributions are interesting.
- I appreciated Section 5 (Other applications). I especially found 5.3 very compelling and relevant to modern ML.
- I found Figure 2 very useful.
- I think that the literature review touches on the key topics, namely the theoretical framework that is used and prior work on the class-dependent analysis.
- The proofs don't have any obvious faults; I haven't scrutinized them deeply.
- The experimental results look good, particularly when including the results in the Appendices. Further, the right column of Figure 3 shows key point that the methods in the paper track the true class-generalization error appropriately.

### Weaknesses
- The writing and presentation can sometimes be unclear. See the comments in the `Requested Changes` section.

---

> ### Author Response · Authors · 2025-05-07
>
> We thank the Reviewer for his positive comments.
>
> > 'Therefore, reasoning only concerning the standard generalization error (red curve) ...'
>
> Thank you for the suggestion. We agree that the revised phrasing improves clarity by explicitly stating that standard generalization errors average across classes. We have updated the sentence accordingly in the revision.
>
> >  Figure 1: Could you add figure titles to the left and right cases?.. use subfigure titles?.. Can the markers be made larger?
>
> Thank you for the helpful suggestions. We have updated the caption of Figure 1 to indicate the number of random seeds used in each experiment. We have also improved the visual quality of the figure to enhance visibility. In addition, we now refer to the left and right panels as subfigures (Figure 1a and Figure 1b) and have added appropriate titles to each for clarity.
>
> > 'this phenomenon using information-theoretic generalization bounds, as they are both data-dependent and algorithm-dependent'...
>
> Thank you. We agree this clarification improves readability. We have adopted the proposed rephrasing in the revision.
>
> > Figure 1: uncertainty as error bars..
>
> We acknowledge the concern. In Figure 1,  we now include error bars  and increased resolution. We believe the updated version is now more readable.
>
> > First paragraph page 2: I am not sure that I fully agree that the second sentence can be concluded from the first... the only difference between the two figures is that a small amount of noise was added; so how can this imply that the learning algorithm affects the generalization?
>
> We appreciate this careful reading. The original statement indeed implied a stronger conclusion than the figure supports. What we meant to highlight is that even small changes in the data, like adding 5% label noise, can cause major shifts in class-wise generalization of several classes, highlighting that there are more factors at play here and further motivating the need to study class-generalization. We’ve revised the paragraph to reflect this more accurately and avoid overstatement.
>
> > Notations: Why are random variables in bold?
>
> We follow a convention from the learning theory and information theory literature where boldface indicates random variables.
>
> > Notations: Can you write a sentence about that the intuition behind the disintegrated conditional mutual information is? ...
>
> In general, “disintegrated conditional mutual information” aims to capture how much information a random variable provides about another, not just globally, but conditioned on finer-grained instances of another variable.  In our context, it enables us to study generalization at the level of individual classes for a specific dataset $Z_[2n]$ realization. Moreover, due to Jensen’s inequality, using disintegrated CMI yields a tighter bound than directly applying standard CMI.  We have added a brief description in the Notation section.
>
> > Related Work: Can a sentence or two be written that explains what Information-theoretic generalization error bounds are? ...
>
> We have added a brief explanation to the beginning of the related work section, stating that information-theoretic generalization bounds were developed to analyze generalization behavior in terms of information shared between the training data and the learned hypothesis, providing insight into how data-dependent and algorithm-dependent interactions govern overfitting. Unlike classical uniform convergence bounds, they are typically tighter in practice.
>
> >  I found it slightly confusing that the notation  Z  exists...
>
> We use the standard notation where Z=(X,Y) denotes a labeled sample pair, and S is the training dataset consisting of n such random samples. Our fine-grained analysis occasionally requires referring to X and Y separately (see Theorem 1); in such cases, we do so explicitly for clarity.
>
> > Based on this section and the Introduction, it is not clear to me why the 5% label noise dataset is significant.. why this level of noise?
>
> Thank you for this valuable question. The main motivation behind the 5% label noise experiment is to demonstrate that even minor corruptions in data distribution can lead to significant disparities in class-wise generalization behavior. We chose this moderate noise level to avoid severely degrading the learning process while still highlighting the sensitivity of class-level generalization. As you can see in the results, these minor corruptions drastically changed the behavior of the class-generalization of certain classes. This highlights the sensitivity of class-wise behavior and motivates the need for further study. We have clarified this in the figure caption and main text.
>
> > Figure 3: Same as Figure 1; please put titles to each sub-figure...
>
> Thank you for this suggestion. We have revised Figure 3 using a grid format with clear titles on both the columns and rows, along with subfigure labels (e.g., Figure 3a, 3b) to improve readability and organization.

---

### Decision · Action_Editor_gwHm · 2025-06-11

**Recommendation:** Accept as is

**Comment:**

The paper introduces the concept of class-generalization error, which reviewers found to be a novel and well-motivated approach to understanding why learning algorithms perform differently across classes. While this is a known issue, it's been an under-studied phenomenon in ML.

Reviewers carefully evaluated the theoretical contributions and proofs, and found them to be rigorous.

One of the things noted by all reviewers was that the theoretical framework extends to other important areas. for example, providing generalization guarantees for sensitive attributes in fairness contexts were seen very promising for future work. In response, the authors strengthened their submission by adding additional experiments on fairness-related tasks.

**Audience:**

As noted by the reviewers, the submission introduces a well-motivated problem, and the insights are of interest to readers working in generalization theory and fairness.

**Claims And Evidence:**

All reviewers agreed that the theoretical contributions in this submission are supported with rigorous proofs, and convincing empirical results on multiple datasets.